# Roadmap for Designing Donor-π-Acceptor Fluorophores in UV-Vis and NIR Regions: Synthesis, Optical Properties and Applications

**DOI:** 10.3390/biom15010119

**Published:** 2025-01-14

**Authors:** Guliz Ersoy, Maged Henary

**Affiliations:** 1Department of Chemistry, Molecular Basis of Disease, Petit Science Center, Georgia State University, 100 Piedmont Avenue SE, Atlanta, GA 30303, USA; gersoyozmen1@gsu.edu; 2Center of Diagnostics and Therapeutics, Georgia State University, 100 Piedmont Avenue SE, Atlanta, GA 30303-5090, USA

**Keywords:** donor acceptor, fluorophores, UV-vis, near-infrared, synthesis, optical properties, applications

## Abstract

Donor acceptor (D-π-A) fluorophores containing a donor unit and an acceptor moiety at each end connected by a conjugated linker gained attention in the last decade due to their conjugated system and ease of tunability. These features make them good candidates for various applications such as bioimaging, photovoltaic devices and nonlinear optical materials. Upon excitation of the D-π-A fluorophore, intramolecular charge transfer (ICT) occurs, and it polarizes the molecule resulting in the ‘push–pull’ system. The emission wavelengths of fluorophores can be altered from UV-vis to NIR region by modifying the donor unit, acceptor moiety and the
π linker between them. The NIR emitting fluorophores with restricted molecular rotations are used in aggregation-induced emission (AIE). D-π-A fluorophores with carboxylic acid and cyano groups are preferred in photovoltaic applications, and fluorophores with large surface area are used for two photon absorbing applications. Herein, we report the synthesis, optical properties, and applications of various D-π-A fluorophores in UV-vis and NIR region.

## 1. Introduction

Donor acceptor (D-π-A) fluorophores with a conjugated π system attracted attention in the last decade. D-π-A fluorophores have a donor group and an acceptor moiety at each end connected by a conjugated π linker system (Figure 1). The π linker system can be a polymethine chain, a heteroatomic or an aromatic molecule [1]. When the D-π-A molecule becomes excited by absorption of a photon, intramolecular charge transfer (ICT) occurs, creating a push–pull system, the donor group being push and the acceptor moiety being pull units. The key to creating a push–pull system is appropriately modifying the molecule. Some examples of these systems do not have a linker unit between donor and acceptor units [2,3]. These scaffolds can be fluorescent due to their conjugated moieties. Similarly, the charge transfer occurs between donor and acceptor units upon photoexcitation, resulting in ICT [4]. However, the linker addition in between can allow a more significant number of modifications to shift the wavelengths to the near-infrared region. Therefore, in this review are focusing on the donor acceptor fluorophores with linker units by presenting various examples.

The NIR window is useful for bioimaging applications since there is less autofluorescence from biomolecules and tissues [5], therefore, D-π-A fluorophores can be modified by changing the donor and acceptor units to have absorbance in NIR. One of the advantages of D-π-A fluorophores is their tunable structure. These fluorophores can be modified by basically three ways: alternation of the donor unit, changing the length of the
π linker and alternation of the acceptor moiety. The length of the
π system is important due to the spatial rearrangement of the fluorophore. Heteroatomic systems like thiophene may improve the polarization and stabilize the conformation of the fluorophore [6]. Some of the donor units used in the literature are: *N*,*N*-dialkylamino [6], *N*,*N*-diphenylamino [6], indoline [7], fluorene [8], carbazole [8] and alkoxy groups [6]. The most used acceptor units are; cyano [6], nitro [6], trifluoromethyl [9], carboxylic acid [6], pentafluorosulfur [9], nitrile [6] and benzothiazole [10]. In addition to these groups, halides can be used as acceptors due to their electron withdrawing properties.

The D-π-A fluorophores are sensitive to their environment, therefore, while measuring the spectroscopic properties, the polarity of the solvent affects the absorption and emission spectra [11]. This property of the molecules is known as solvatochromism [12]. The interaction between the fluorophore and solvent affects the ICT and shifts the absorption spectra, as well as changing the Stokes shift [13,14]. It is favorable to have a larger Stokes shift separating absorbance and fluorescence wavelengths and dyes with extended conjugation systems were able to reach that [14]. It has been reported that both dipole-dipole interactions and hydrogen bonding can promote the bathochromic shift of emission spectra by stabilizing the excited energy level of the fluorophore [13]. Rigidity of the fluorophore is another subject that is addressed by researchers discussing how the more rigid and conjugated the structure is, the brighter and more red shifted absorbance can be obtained [15,16].

This review specifically focuses on D-π-A fluorophores with various moieties and geometries published after 2014.

## 2. D-π-A Fluorophores in the UV-Vis Region

The majority of D-π-A fluorophores reported so far are in the UV-vis region. The synthesis and photochemical properties of fluorophores in the UV-vis region with different donor and acceptor units, and
π linkers, will be discussed in this section.

### 2.1. Pentafluorosulfonyl (SF_5_) Acceptor Group

It is known that trifluoromethyl (CF_3_) is a strong electron withdrawing group, on the other hand, it was reported that the SF_5_ group has much stronger electron withdrawing effect. W. Sheppard reported the synthesis of arylsulfur pentafluorides [17] and studied the electron withdrawing effect of the SF_5_ group which showed a greater inductive effect than CF_3_ [18]. Due to this strong electron withdrawing effect, Gautam et al. used pentafluorosulfonyl (SF_5_) as an acceptor group for the synthesis of push–pull fluorophores [9]. In their study, they synthesized dyes with diphenylamino as the donor unit and SF_5_ as the acceptor unit connected with different
π systems via Pd catalyzed coupling reactions [9]. The general scaffold of D-π-A system is schematically represented in Figure 2.

The structures of synthesized dyes from **1a**–**f** with different
π units are shown in Figure 3. Fluorophore **1a** was synthesized via Suzuki coupling, **1b** and **1e** were synthesized via Heck coupling, and **1c**, **1d** and **1f** were synthesized via the Sonogashira reaction. As an example, the synthesis of **1a**, **1c** and **1e** were shown in Figure 1. The syntheses for **1a**–**1e** start from bromo(pentafluorosulfanyl)benzene and couple with different precursors via Pd catalysts. The photophysical properties of dyes were investigated in dichloromethane (DCM), toluene and acetone. The dyes **1a**–**f** showed absorbance maxima between 344 and 407 nm, and had fluorescence maxima between 449 and 506 nm. Compounds **1d**–**f** showed a bathochromic shift which is a result of ICT in their longer
π system. The quantum efficiencies of **1d**–**f**, 0.40, 0.23 and 0.48 were significantly higher than **1a**–**c** which were 0.016, 0.20 and 0.18, respectively [9].

Time-dependent density functional theory (TD-DFT) studies showed that the HOMO was located at the diphenylamino donor group and LUMO was located at the SF_5_ acceptor moiety. As presented in Figure 4, the band gap decreases from 3.80 eV to 2.81 eV for fluorophores **1d**–**f** when compared with fluorophores **1a**–**c**. In the case of **1f**, the addition of the thiophene unit lowered both HOMO and LUMO levels with a band gap of 2.84 eV. This study shows that increasing the conjugation and adding an heteroatomic unit to the
π linker system decreases the band gap and results in a bathochromic shift. Although SF_5_ is seen as a good electron acceptor unit, the fluorescence signals were in the UV-vis region [9].

### 2.2. Diazines

Diazines are heterocyclic dinitrogen atoms containing aromatic molecules [19]. Two nitrogen atoms in the diazine structure do not contribute to the aromaticity of the molecule so the structure acts as a relatively electron-deficient system [20]. In particular, pyridazine and pyrimidine are used as acceptor units in D-π-A molecules due to their aromaticity and enhancing intramolecular charge transfer characteristics [21]. An example of using diazines as acceptor units with an extended linker unit was reported by Achelle et al. [22]. The diazine containing chromophores **5a**–**e**, **6a**–**e** and **7a**–**e** with long
π systems were synthesized via the Sonogashira cross-coupling reaction for nonlinear optical (NLO) applications [22]. The fluorophores **5** and **7** both contain a pyrimidine acceptor unit, however, they differ in the linker unit as phenyl and thienyl rings, respectively. Fluorophore **6** has a quinoxaline acceptor and a linker unit containing a phenyl ring. The synthesis of these fluorophores with a diazine acceptor unit and different R groups as donor units such as phenyl, methoxyphenyl, dimethylaminophenyl, diphenylaminophenyl, trimethylsilane and hydrogen are given in Figure 2. The starting compounds **2**–**4** were synthesized by the condensation of 4-methylpyrimidine or 2-quinoaxaline, an aldehyde under basic conditio and Aliquat 336 (phase transfer catalyst) to yield the E-conformation as reported by Vanden Eynde earlier [23]. Compounds **5**–**7f** were synthesized by cleaving the trimethylsilane group on **5**–**7e** by potassium hydroxide in methanol [22].

The optical studies of the derivatives from molecules **5**–**7** were measured in DCM showing absorbance wavelength maxima around 312–419 nm, and emission spectra between 420 and 630 nm in the UV-vis region. The compounds have large Stokes shifts, and they did not show E–Z isomerization. When the two different diazine units, pyrimidine (**5**) and quinoxaline (**6**), were compared, **5a**–**d** had
λ_em_ = 420–612 nm and **6a**–**d** had
λ_em_ = 447–647 nm, resulting in a red shift for fluorophore **6**. The thienyl ring containing
π linker **7a**–**d** had
λ_em_ = 462–607 nm. For molecules containing pyrimidine and quinoxaline acceptor units, dimethylaminophenyl (**c**) and diphenylaminophenyl (**d**) were the most bathochromic shifted. In the case of the pyrimidine acceptor with the thienyl unit, **7c** showed the highest quantum yield
Φ = 0.80. Fluorophore **5c** (Φ = 0.54) showed a higher quantum yield than **7c** (Φ = 0.07). The investigation of photophysical properties of the compounds in aprotic solvents showed the solvatochromic behavior of the fluorophores. With increasing solvent polarity, a bathochromic shift was observed for the emission band while the absorption band did not show significant changes [22]. The benzene ring consisting of one more nitrogen atom, which is called triazine, can also be used to make donor acceptor scaffolds and can find applications as viscosity sensors [24].

### 2.3. Thiophene and Thienothiophenes

Thiophenes are heterocycles similar to cyclopentadiene which contain a sulfur atom, and the lone pair on the sulfur atom contributes to the aromaticity of the ring since the p orbital on sulfur is conjugated to the p orbitals of the heterocycle. Thiophene heterocycles can be used as donor or linker units. An example with thiophene donor units was reported by Peng et al. where two different systems contain triazine and benzene cores, and both systems showed absorbance in the UV-vis region [25]. Balakirev et al. reported donor acceptor fluorophores with a thiophene linker unit where the scaffold has a triphenyl amine donor and several acceptor units [26]. Arahchige et al. reported a thiophene linker containing fluorophores with quinolinium units which has emission in the NIR region upon being reduced with a NAD(P)H addition [27].

In 2016, Manuela et al. reported the synthesis of thienothiophene containing fluorophores using two different sets of Suzuki cross-coupling reactions, which are given in Figure 3. TT is used as the π linker, dicyanovinyl and thiobarbituric acid groups are used as acceptor units, and various donor units were compared according to their optical properties. While compounds **8a**–**d** containing only the donor unit with the aldehyde group at the other end had λ_em_ = 413–545 nm, fluorophores **9a**–**d** and **10a**, **b**, **d** showed a bathochromic shift due to the addition of strong acceptor units dicyano and thiobarbituric acid, instead of the aldehyde group. Compound **10d** specifically had fluorescence in the NIR region as λ_em_ = 710 nm [28].

Thienothiophene (TT) has an aromatic structure consisting of two fused thiophene rings. Thionothiophenes have four isomers: thieno[3,2,*b*]thiophene **11** in Figure 4, thieno[2,3,*b*]thiophene **12** in Figure 4, thieno[3,4,*b*]thiophene and thieno[3,4,*c*]thiophene [29]. The first two isomers are used as the examples given in this section. The conjugated and rigid structure of thienothiophenes make them good candidates as linker units in donor acceptor fluorophores.

**Scheme 3 biomolecules-15-00119-sch003:**
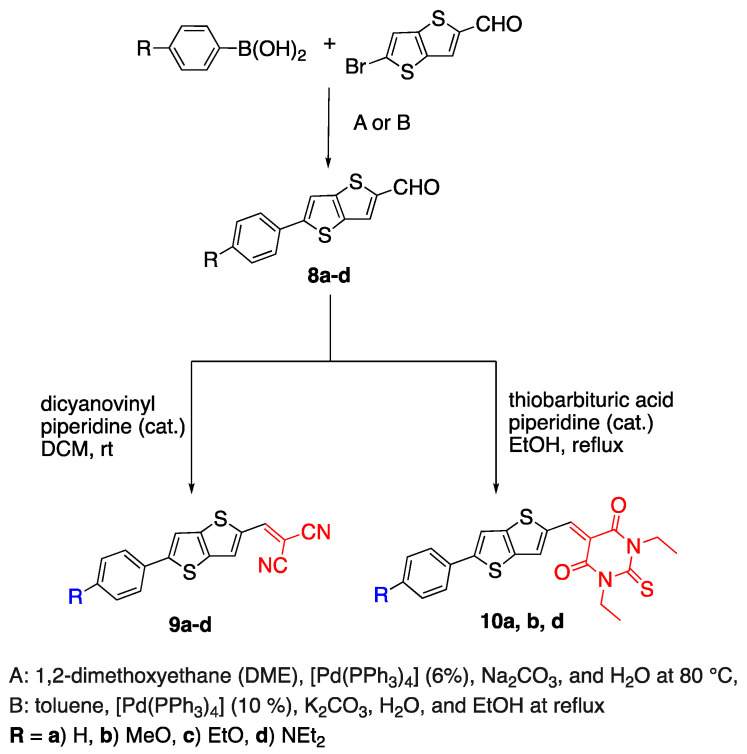
Synthesis of chromophores with TT linker system.

Thienothiophenes also have an electron-donating character due to the lone pair electrons on sulfur atoms [29]. Podlesny et al. used thieno[2,3,*b*]thiphene **11** and thieno[3,2,*b*]thiophene **12** as donor units and investigated their properties with various acceptor units. The synthesis of the fluorophores with some of the selected acceptor units such as indan-1,3-dione (**a**), *N*,*N*-diethylthiobarbituaric acid (**b**), ThDione (**c**) and malononitrile (**d**) are presented in Figure 4 [30]. The first step of synthesizing fluorophores **14** and **16** was the Vilsmeier Haack reaction by reacting POCl_3_ with DMF followed by the dropwise addition of **11** or **12**. The last step of the synthesis was Knoevenagel condensation with Al_2_O_3_ in DMF to yield compounds **14** and **16**. In another study, Podlesny et al. used thieno[3,2,*b*]thiophene (**12**) as a donor unit and reported the synthesis of D-π-A fluorophores with different acceptor units and a butadiene linker in between, which makes the fluorophore undergo E/Z isomerization upon light (Figure 4) [31]. The synthesis of compound **17** was not successful via the Vilsmeier Haack reaction, therefore it was synthesized by 3-(*N*,*N* dimethylamino) acrolein with LDA. The second step was the Knoevenagel condensation to yield chromophore **18** E/Z isomers in the light and only E isomer when performed in the dark. The E/Z isomers were characterized by ^1^H and ^13^C NMR spectroscopies, and it was concluded that the photoisomerization was affected by the acceptor unit [31].

The absorbance spectra of compounds **14a**–**d** and **16a**–**d** are measured in *N*,*N*-dimethylformamide (DMF). The absorption maxima of fluorophores **14a**–**d** ranges from 383 to 444 nm, whereas the absorption maxima of fluorophore **16a**–**d** are in the range of 389–446 nm, with the lowest
λ_max_ for compounds **14d** and **16d** with the malononitrile acceptor, and the highest
λ_max_ for fluorophores **14b** and **16b** with the barbituric acid acceptor units [30]. The absorbance maxima for **18a**–**d** was reported as being between 428 and 479 nm in DMF, showing a bathochromic shift upon the addition of an additional HC=CH bond specifically for **18a** and **18b**. The DFT studies of the chromophores show the largest band gap exists for **14b**, **16b** and **14d;**
**16d** have a smaller band gap, and compounds **18a**–**d** have a larger band gap which is observed for E isomers [31].

**Scheme 4 biomolecules-15-00119-sch004:**
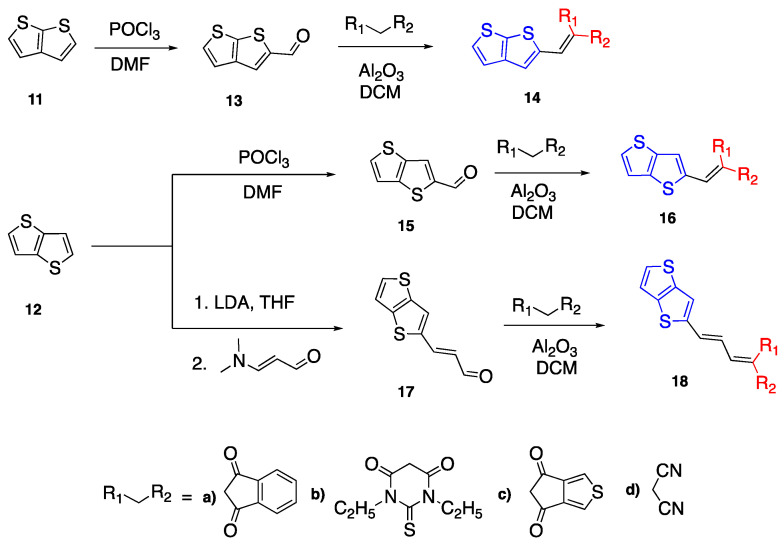
Synthesis of fluorophores using TT as donor group.

### 2.4. Nonlinear D-π-A Fluorophores

In addition to the linear D-π-A-shaped fluorophores, different types of scaffold can be designed, such as tripolar (D(-π-A)_3_) and quadrupolar (D-π-A-
π-D) structures, which are illustrated in Figure 5 [6]. The conjugation in these systems is higher than that in the linear D-π-A fluorophores since they have three donor or acceptor groups. They are good candidates for multiphoton absorption (MPA) due to their high conjugated system. Here, two different examples will be discussed, D-π-A fluorophores with triphenylamine and a pyrimidine core.

**Figure 5 biomolecules-15-00119-f005:**
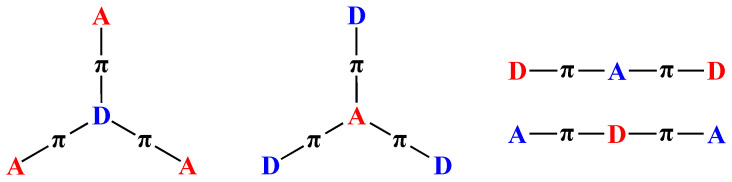
Tripodal and quadrupolar donor acceptor structures.

#### 2.4.1. Triphenylamine Core

Tripodal, also known as star-shaped D(-π-A)_3_ and (D-π-)_3_A structures, are examples of different spatially arranged D-π-A fluorophores. In recent years, triphenylamine containing fluorophores have been synthesized, which have triphenylamine in the middle and different groups attached by the
π linkers [8,32,33]. Triphenylamine is preferred for the synthesis of tripolar structures since it has C_3_ symmetry, a coplanar structure and an efficient synthesis of the molecule with different groups [34]. Cvejn et al. synthesized star-shaped fluorophores using triphenylamine as the core unit in the middle [34]. The syntheses of different fluorophores were followed by Suzuki-Miyoura, Sonogashira and Heck-Mizoroki reactions [34]. The syntheses of the core structures are shown in Figure 5.

Triiodo compound **20a** was synthesized from commercially available triphenylamine (**19**) in two ways, as reported by Cvejn et al. The first method reported using the HgO/I_2_ system which resulted in a high yield but low selectivity towards mono- and bisubstituted products. Therefore, as a second method, they carried out the synthesis with the AgNO_3_/I_2_/EtOH system, which yielded the desired products with reasonable yields. Compound **20b** was synthesized from tris(4-iodophenyl) amine, trimethylsilylacetylene (TMSA), tetrabutylammonium fluoride (TBAF) and Pd catalyst by the threefold Sonogashira cross-coupling reaction. Then, starting from **20a** and propargyl alcohol, fluorophore **20c** was synthesized via another threefold Sonogashira coupling, and **20d** was synthesized by the oxidation of fluorophore **20c** by the Dess Martin periodinane reagent [33,34].

Fluorophores **5**–**7** reported earlier in Figure 2 were reacted with tris(4-ethynylphenylamine) **(20b)** starting with Pd (0) precatalysts, diisopropylamine and 1,4 dioxane via three-times Sonogashira coupling at 92 °C to yield tripolar fluorophores **21**–**24** given in Figure 6. Fluorophore **25,** which has a pyrazine ring, was synthesized by a different method using potassium hydroxide in DMSO since the yield was low for the method used for **21**–**24** due to the low acidity of the methyl hydrogens [32]. The compounds **21**–**25** have an emission between 555 and 630 nm with high quantum yields
Φ = 0.37–0.67. Fluorophores **23** and **24** have a bathochromic shift which may be due to the higher
π conjugation and rigidity of the quinoxaline ring. The addition of the thienylene ring instead of phenylene resulted in a bathochromic shift for compound **22** but a blue shift for compound **24** which has a quinoxaline ring, which may arise due to the disruption of the coplanar structure in fluorophore **24** [32].

In another study reported by Cvejn et al., fluorophores with mono- and dicyano groups (**26**–**29**) were synthesized via a threefold coupling reaction [34]. Fluorophore **26** was synthesized from (4-cyanophenyl) boronic acid and **20a** via the Suzuki coupling reaction. Fluorophore **27** with dicyanovinyl acceptor was synthesized via Al_2_O_3_ catalyzed Knoevenagel condensation reacting malononitrile and **20e** with a high yield. Fluorophores **28** and **29** were synthesized via the Sonogashira cross-coupling reaction starting from **20b**, 2-bromoisophthalonitrile and 2-iodothiophene-3,4-dicarbonitrile, respectively. The fluorophores **26**–**29** have an absorption between 448 and 611 nm, fluorophore **27** having the one with the highest bathochromic shift. The highest quantum yields observed for fluorophores **26** and **28** were 0.64 and 0.69, respectively [34].

#### 2.4.2. Pyrimidine Core

In addition to the triphenylamine ring, there are examples of a pyrimidine ring used as the core of an octupolar structure [35]. The electron-deficient character of pyrimidine also makes it useful as an acceptor unit. The synthesis of 2-ethynyl-4,6-bis(arylvinyl)pyrimidines was reported by Martin et al. [35] and given in Figure 6. The 2-iodo-4,6-dimethylpyrimidine **30** was reacted with an aldehyde which has a donor group represented as D: dimethylamino (**a**), dibutylamino (**b**) and methoxy (**c**) groups to yield **31a**–**c**. Fluorophores **32a**–**c** were synthesized from **31a**–**c** by two steps, shown as i and ii in Figure 6. The optical studies of fluorophores were conducted in DCM and Stokes shifts were between 93 and 105 nm. Fluorophores **32a**–**c** had absorbance maxima of 440, 455 and 364 nm, and emission maxima of 545, 547 and 457 nm, respectively. Fluorophore **32c**, which had methoxy as the donor group, showed the least quantum efficiency at 0.02, while the quantum efficiency of **32a** and **32b** were 0.47 and 0.52 [35]. The triple bond in the linker unit was preferred since it gave a coplanar structure and increased stability.

Kournotuas et al. studied the photophysical properties of the pyrimidine acceptor containing fluorophores **33a**,**b**–**37a**,**b** with dimethylamino (**a**) and diphenylamino (**b**) donors [36]. Fluorophores **33a**,**b** and **34a**,**b** were synthesized from 4-methyl pyrimidine and benzaldehyde (1 and 2 equivalence, respectively) with a donor unit, using Aliquat 366 catalyst under basic conditions, as presented in Figure 7 [37,38].

The synthesis of **37a**,**b** was reported using a three-step synthetic route, presented in Figure 8, by Feckova et al. [39]. The first step is C4 regioselective Suzuki-Miyaura coupling with styrylboronic acid, starting from 2,4-dichloro-6-methyl pyrimidine, to yield **35a**,**b**. This step is followed by another Suzuki-Miyaura coupling at the C2 position. The molecular optimizations showed that the molecules have a planar backbone, whereas the diphenylamino group has a propeller shape. The photophysical studies were performed in DCM, absorbances were between 392 and 426 nm, and the dyes had an emission between 493 and 538 nm. The highest bathochromic shift and quantum efficiencies were seen for dyes **34b** and **37b** with the diphenylamino group. The fluorophores showed solvatochromic properties and large Stokes shifts in various solvents.

### 2.5. Fluorene as Linker Unit

Fluorene is another molecule that is used for the synthesis of D-π-A fluorophores. Fluorenes are highly conjugated, rigid and have a planar structure promoting red shifted absorbance. Shaya and coworkers reported the synthesis of 2,7-disubstituted fluorene containing fluorophores and studied their optical properties. Dimethylamine and several heterocycles were used as the donor units, and various acceptor units were studied [40]. The general structure of fluorene containing fluorophores is shown in Figure 7.

The synthesis of fluorene containing fluorophores and their different acceptor groups are outlined in Figure 9. The first step of the synthesis is the dimethylation of the carbon at position 9. The second step is the regioselective mononitration of C7 followed by the reduction with iron and ammonium chloride to afford the amino group. This is followed by reductive amination using NaCNBH_3_ and paraformaldehyde to obtain compound **40**, which contains the dimethylamino group as a donor and different groups as acceptors. Compound **40a** with an aldehyde unit was synthesized via formylation with n-BuLi. Compound **40b** was synthesized in two steps starting with the Grignard reaction for adding the ethynyl group, and this followed by oxidation with MnO_2_ to obtain the ketone structure with a high yield. Knoevenagel condensation was carried out to synthesize **40b** by methylenemalonitrile and Al_2_O_3_. After the generation of stannyl fluorene in situ, **40c** was obtained by coupling it with the benzothiadiazole ring. For **40d** that was peptide functionalized, first the succinic linker was attached to the R position by the Grignard reaction of **40** and succinyl anhydride in THF at 60 °C, then this was followed by the ligation reaction of n-butyl amine and two activators, *N*,*N*-diisopropylcarbodiimide and I-hydroxybenzotriazole [40].

The spectroscopic studies of fluorophores **40a**–**d** were performed in various solvents such as water, ethanol, acetonitrile, DMSO, chloroform, dioxane and toluene. In water, all fluorophores had emissions above 600 nm, and fluorophore **40b** showed the highest red shift at 682 nm. In DMSO, ethanol and acetonitrile **40b** and **d** showed a bathochromic shift, and for **40b** the emission was in NIR region in DMSO as
λ_em_ = 710 nm. The ethynyl group containing **40a** had a significant bathochromic shift and the most significant difference was in acetonitrile from 521 nm to 620 nm. The addition of a benzothiadiazole ring showed a blue shift of the fluorophore **40c**, which had an emission only in aprotic and nonpolar solvents [40].

Another example of fluorophores with a fluorene linker is presented in Figure 10. The synthesis of fluorophore **43**, which has a diphenylamine donor and benzothiazole acceptor units, was reported by Kannan et al. via a Pd-catalyzed amination reaction starting from 7-bromo-9,9-diethyl-9*H*-fluorene-2-carbaldehyde **42** [41]. First, the carbaldehyde **41** was reacted with 2-aminobenzenethiol in DMSO to form the precursor compound **42**. Then, the copper-catalyzed reaction of **42** with diphenylamine in the presence of 18-crown-6 and potassium carbonate yielded fluorophore **43**. The absorbance spectra of **43** showed absorbance and emission maxima at 392 and 475 nm, respectively in THF.

Kannan et al. also reported the synthesis of diphenylamine donor fluorenes with different type of acceptors such as 2-benzothiazolyl, benzoyl, 2-(4-pyridyl)-vinyl, 2-benzoxazolyl, 2-quinoxalinyl, 2-(*N*-phenylbenzimidazolyl and 2-(4,5-diphenylimidazolyl) [41]. The synthesis of fluorophore **45** with dibenzothiazole units was reported by Belfield et al. starting from 9,9-diethyl-9*H*-fluorene-2,7-dicarbaldehyde **44** and 2-(tri-n-butylstannyl)-benzothiazole via Pd catalyzed Stille coupling [42]. The absorbance spectra of **45** in THF showed three peaks with maxima at 358, 375 and 398 nm [43]. The synthesis of fluorophore **47** with diphenylamino groups was reported by Stewart et al. via a Pd catalyzed reaction [43]. The absorbance and emission of fluorophore **47** were 380 and 396 nm, respectively, in THF. The fluorophores **43** and **47** showed large Stokes shifts in polar and nonpolar solvents. The quantum yields of fluorophores **43**, **45** and **47** were between 42 and 80% in various solvents [10].

### 2.6. Carbazoles as Linker and Acceptor Unit

Carbazoles are another example of moieties commonly used in D-π-A fluorophores. Similar to fluorenes, they also have a planar structure. The difference between carbazoles and fluorenes is the nitrogen atom which makes the structure more electron-rich and conjugated [44]. Carbazoles can be modified at 3 and 6 positions to give mono- and disubstituted products. As an example, Raikwar et al. synthesized carbazole containing fluorophores with D_1_-π-A-π-D_2_ structures, as outlined in Figure 11 [45]. In this system, D_1_ is the carbazole unit, the acceptor is the dicyano vinylene and D_2_ represents three different donor units. The first step to synthesize 1-(9-ethyl-9*H*-carbazol-3-yl) ethan-1-one **49** was reported by R. Tang and W. Zhang earlier [46]. They synthesized monosubstituted carbazole with Friedel-Crafts acetylation using ZnCl_2_ as the catalyst; while AlCl_3_ gave both mono- and disubstituted products, ZnCl_2_ gave monosubstituted as the main product. The difference arises from the high acidity of AlCl_3_ and the mild acidity of ZnCl_2_. Then, intermediate **49** was reacted with malononitrile undergoing Knoevenagel condensation to give compound **50**. Finally, the malononitrile functionalized carbazole reacted with the commercial aldehydes to yield fluorophores **51a**–**c** [46].

The absorption and emission studies of the three fluorophores with different D_2_ units were conducted in CHCl_3_. Fluorophores **51a**–**c** have absorption maxima at 492, 494 and 532 nm, respectively. According to the studies, structures of **51a** and **51b** were not planar because of the free rotation around the double bond connecting D_2_ to the carbazole structure. On the other hand, compound **51c** is planar and rigid, with a better
π conjugation, which explains the red shift in absorbance. The emissions of fluorophores **51a**–**c** were 570 nm, 615 nm and 603 nm, respectively. Fluorophore **51b** has the most bathochromic shifted emission, and this may be due to the stronger electron donating effect of the diphenyl group than the julolidine group. The effect of solvent polarity on the absorbance of dyes was also investigated. Fluorophores **51a** and **b** were not affected by the polarity of the solvent, whereas **51c** was affected. DFT studies were conducted to find the optimized geometries of the molecules, and their bond lengths and strengths. According to these studies, fluorophores showed a shorter bond length in polar solvents. The geometric optimization of the structures showed that the D_2_ unit is planar with the acceptor unit, but the D_1_ unit is not planar to them. While the angle between the D_2_ and the acceptor units is 4–6°, the angle between the D_1_ and acceptor units is 39–42°, which shows that the D_1_ unit is not planar, therefore, the HOMO is on D_2_ and LUMO is on the acceptor [45].

Another set of fluorophores with a carbazole donor, oligothiophene linker and cyanoacetic acid acceptor were reported by Wang et al. [47]. Later, Sutton et al. reported the synthesis of fluorophores **55**–**57,** which have a modified length of thiophene
π bridge [48]. They investigated the effect of distance between donor and acceptor groups by changing the length of the thiophene linker.

The first set of fluorophores **55**–**57** were synthesized by Wittig condensation starting from triphenylmethyl(9-octyl-carbazole-3-yl) phosphonium bromide **52** and 2,5-dicarbaldehyde thiophene and 1,8-diazobicyclo[5.4.0]undec-7-ene (DBU), as illustrated in Figure 12 [48]. This process was followed by a reaction with trimethyloxonium tetrafluoroborate in dry diethyl ether and ethyl diisopropyl amine to synthesize fluorophores **55**–**57**. The R group on the acceptor corresponds to the OH and OCH_3_ groups for fluorophores **55**–**57a** and **55**–**57b**, respectively.

The second set of fluorophores **60** and **61** have a monothiophene bridge between the donor and acceptor units. Fluorophores were synthesized by the Suzuki coupling reaction starting from 3-iodo-9-octylcarbazole **58** and 5-formylthienyl boronic acid catalyzed by Pd (PPh_3_)_4_ in THF under an argon atmosphere by 4-h reflux, as shown in Figure 13. This was followed by Knoevenagel condensation of the aldehyde with cyanoacetic acid. Fluorophores **60** and **61** were synthesized according to the same Knoevenagel condensation and cyanoacrylic acid esterification procedures, as reported earlier [48]. The absorption of the fluorophores was between 470 and 485 nm, and the bathochromic shift was observed for four and six thiophene containing bridges. To understand the effect of the bridge length, the photophysical properties and TD-DFT studies were performed. According to the electronic absorption studies, increasing the bridge length promotes a bathochromic shift of
λ_max_ for both **a** and **b** compounds (Figure 12). When bithiophene and monothiophene are compared, there is a 0.11 eV shift. However, when one more thiophene unit is added, for terthiophene the shift is 0.01 eV. This difference in the shift is reported due to the rotation around the thiophene bridge which may have disturbed the conjugation. However, for this structure, while bithiophene in closer to planarity, terthiophene may be disturbed planarity. According to the calculated ground state conformations, the **56a** and **56b** prefer the anti- and **57a** and **57b** prefer anti-anti conformations. Using TD-DFT studies, they reported that when the fluorophores are excited, ICT occurs from carbazole to the acceptor unit through the thiophene bridge. The increase in bridge length decreases the electron density transferred from carbazole to the acceptor, while increasing the dihedral angle from **55a**,**b** to **57a**,**b** promotes ICT. The solvent effect on the photophysical properties of dyes was investigated using different solvents, and while the **a** compound showed high solvatochromism, the **b** compound was less affected. When the solvatochromic behavior of **55a** was investigated, it was reported that the deprotonation of the cyanocarboxylate unit may be causing this unusual behavior. The emission studies showed a small difference in the
λ_em_ of **61.** On the other hand, **56a**, **57a** and **b** showed a high bathochromic shift as
λ_em_ = 703, 715 and 701 nm, respectively [48].

Another set of fluorophores with carbazoles was reported by Zheng et al. [49]. The carbazole group in fluorophores **63** and **64** act as a donor unit, and the acceptor units are pyrylium and quinolinium salts, respectively. As presented in Figure 14, the fluorophores were synthesized via Knoevenagel condensation between the carbaldehyde, pyrylium and quinolinium salts.

As shown in Figure 15, fluorophores **68** and **69** have diphenylamino as a donor unit and carbazole is used as the
π linker unit. 2-carbaldehyde-7-bromo carbazole **66** was synthesized from 2,7-dibromo-9-ethyl-9H-carbazole **65** and n-BuLi in THF and DMF. After the formation of carbaldehyde, it was reacted with diphenylamine, Cs_2_CO_3_ and Pd (OAc)_2_ in toluene. Finally, diphenylamino carbazldehyde was reacted with the pyridinium and quinolinium salts under basic conditions to yield fluorophores **68** and **69**.

The absorbance maximum of fluorophore **63** was reported as 398 nm in DMSO. Upon the change of acceptor unit from the pyridinium to quinolinium group in fluorophore **64**, the absorbance maxima shifted to 448 nm. The fluorophores **68** and **69** with diphenylamine donor group showed a red shift, and the absorbance maxima were reported as 458 nm and 507 nm, respectively. The data showed that stronger donor and acceptor groups enhance ICT and promote red shift. The calculated band gaps using DFT studies were reported as 2.18 eV and 2.04 for fluorophores **63** and **64**, and the band gap decreased upon the addition of the diphenylamine group as 1.57 and 1.43 for fluorophores **68** and **69**.

Brzeczek et al. reported another example of carbazole containing D-π-A fluorophores which have a star-shaped structure. The synthesis of the star-shaped 1,3,5-tricarbazolebenzenes was performed via Suzuki coupling reactions [50]. The synthesis of the carbazole units of the star-shaped fluorophores is shown in Figure 16. The first step is the iodination of the carbazole to give monoiodinated and diiodinated carbazoles followed by the addition of octyl bromide to yield compounds **71** and **73**. The thienyl group was added to **73** by 2-(tributylstannyl) thiophene, cesium fluoride, copper iodide and Pd (PPh_3_)_4_ under argon atmosphere to synthesize **74**. Then, compounds **71** and **74** were dissolved in THF, and n-BuLi was added dropwise at −78 °C. Finally, 2-isoproxy-4,4,5,5-tetramethyl-1,3,2-dioxaboralane was added to yield the desired boronated products **72** and **75** [50].

The star-shaped fluorophores with carbazole moieties were synthesized as outlined in Figure 17 by two different methods, i and ii. Fluorophores **79a**–**c** were synthesized starting from **76** or **77** and 1,3,5-tribromobenzene or 1,3,5-tribromomethoxybenzene by dissolving in THF and water, followed by the addition of potassium carbonate and bis-(triphenylphosphine)palladium (II) dichloride [50]. The absorption maxima of **79a**–**c** were reported as 292 nm for **79a**, 286 nm for **79b** and 309 nm for **79c** in DCM. While the methoxy group did not show a significant effect, fluorophore **79c** showed a bathochromic shift upon the addition of a thienyl ring, which may be due to the increased conjugation.

### 2.7. Tetracyanobutadienyl Acceptor Group

As discussed earlier, the cyano group is a good electron acceptor, and a tetracyano acceptor containing dyes is used for photovoltaic applications. The synthesis of a tetracyano acceptor containing donor acceptor fluorophores was reported earlier [51,52,53]. As an example, Michinobu et al. reported the synthesis of D-π-A dyes with different donor groups and a tetracyano acceptor group [54]. The synthesis of fluorophores **83a**,**b** and **86a**,**b** is presented in Figure 18. The reaction is reported to proceed as a [2 + 2] cycloaddition between the exocyclic double bond of the tetracyano ethylene **80** or tetracyanoquinodimethane (TCNQ) and the triple bond of the alkyne **81**. Then, the intermediates **82** and **85** are formed and, due to the ring strength of cyclobutene, the reaction yields **83a**,**b** and **86a**,**b** as a concerted mechanism [52]. The absorbance maxima of fluorophores in DCM were reported as 570 nm, 759 nm, 600 nm and 750 nm for **83a**, **83b**, **86a** and **86b**, respectively.

### 2.8. Cyanoacetic Acid Acceptor Group

Cyanoacetic acid is another commonly used acceptor group in D-π-A fluorophores [55,56,57,58]. Quinoxaline was used as an acceptor group in dye **6** (Figure 2) previously, and due to its conjugated structure, it can also be used as a linker. Yang et al. reported the synthesis of two quinoxaline D-π-A fluorophores with bulky donor groups, and cyanoacetic acid as an acceptor unit, as shown in Figure 19 [7]. The fluorophores **90a** and **90b** were synthesized starting with the Suzuki coupling reaction of 5,8-dibromo-2,3-diphenylquinoxaline **87** and (5-formylthiophen-2-yl) boronic acid. The thiophene ring increases conjugation while it also contributes to the acceptor unit. This is followed by the addition of an indoline donor unit via Suzuki coupling reactions to yield compound **89**. Finally, a cyanoacetic acid acceptor was added via Knoevenagel condensation to yield the desired products **90a**,**b**. The absorbance spectra of fluorophores showed two peak maxima where the absorption band was between 360 and 420 nm and corresponds to the quinoxaline linker unit. The absorption maxima of dyes **90a** and **90b** with bulkier donor units were 522 nm and 534 nm in DCM [7].

### 2.9. 3,4-Ethylenedioxythiophene (EDOT) as Linker Unit

3,4-Ethylenedioxythiophene (EDOT) is an important strong donor group that is used in small-molecule solar cells [59,60]. It was reported that, due to the strong electron donating ability of EDOT, the HOMO level increases [61]. Demeter et al. reported that fluorophores with an EDOT unit can be used together with the dicyanovinyl acceptor group, and when multiple EDOT units were used, bathochromic shifts were observed [59]. Antwi et al. reported the synthesis of A-D-A-type fluorophores with an EDOT donor unit and three different acceptor units: 1,3-indanedione, 3-ethyl rhodanine and ethyl cyanoacetate [62]. The synthesis and the resulting fluorophore structures are shown in Figure 20. First, 5,5′-(2,3-dihydrothieno[3,4-*b*][1,4]dioxine-5,7-diyl) bis(4-hexylthiophene-2-carbaldehyde) **92** was synthesized by activating the alpha positions of EDOT. Then, the fluorophores **93**–**95** were synthesized by Knoevenagel condensation in chloroform under basic conditions. The absorption maxima of fluorophores **93**–**95** were 570 nm, 540 nm and 510 nm in solution, and 592 nm, 550 nm and 513 nm in solid state, respectively. The lowest band gap was reported for **93** with the 1,3-indanedione acceptor group. Since EDOT increases the planarity of the structure, this may result in the aggregation and stacking of the fluorophores in solid form, which is also seen in the decreasing band gap of fluorophores in solid state [62].

## 3. D-π-A Fluorophores in NIR Region

So far, the synthesis and optical properties of D-π-A fluorophores in the UV-vis region have been discussed. As mentioned before, fluorophores in the NIR region have advantages over those in the UV-vis region, such as a longer wavelength penetrating deeper into the tissue and not overlapping with the emission of biological tissues [63]. Due to these reasons, there is an ongoing interest in the fluorophores in the NIR region, and in this section the synthesis and optical properties of some of the selected D-π-A fluorophores will be discussed.

### 3.1. 4,5,5-Trimethyl-2,5-dihydrofuran (TCF) Acceptor

4,5,5-Trimethyl-2,5-dihydrofuran (TCF) is a strong acceptor unit used in D-π-A fluorophores [64,65,66]. Remond et al. synthesized fluorophores using 2-dicyanomethylidene-3-cyano-4,5,5-trimethyl-2,5-di-hydrofuran (TCF) as an acceptor and modified at in three positions with extra electron withdrawing phenylthio or phenylsulfonyl moieties [1]. They synthesized compounds **97**–**99** by the Knoevenagel reaction and then combined them with TCF in a microwave to synthesize the final products, as outlined in Figure 21. The fluorophores showed large Stokes shifts and the emission maxima were shifted to NIR. Fluorophores **97**–**99** had absorption maxima at 570 nm, 509 nm and 580 nm, and fluorescence maxima at 788 nm, 718 nm and 790 nm, respectively, in chloroform. Fluorophores **97** with the phenylthiol group and **99** with the TCF acceptor showed the highest red shift and Stokes shifts of 218 nm and 210 nm. The solvatochromic properties of the fluorophores were further investigated in seven different solvents, namely cyclohexane, toluene, chloroform, dichloromethane, tetrahydrofuran, acetone and dimethylsulfoxide. The solvent polarity versus Stokes shift of the molecules were investigated using the Lippert–Mataga plot, and the large dipole moment changes were reported.

### 3.2. Pentafluorosulfonyl and Tetracyano Acceptor

As discussed earlier, fluorophores containing a SF_5_ acceptor moiety (Figure 1) showed emission in UV-vis. However, with the further functionalization of **1c**, two more compounds were synthesized as given in Figure 22, and their emission was red shifted. Compound **100** was synthesized from **1c** and tetracyanoethylene (TCNE). The synthesis of **101** was a slow ring opening reaction of tetracyanoquinodimethane (TCNQ) and further purification was necessary since the starting materials were not consumed totally. The absorbances of **100** and **101** were measured in DCM and reported as 485 nm and 667 nm. The red shifted absorbance properties can be explained by the electron withdrawing properties of tetracyano units.

As the combination of SF_5_ and tetracyano moieties, they make a strong acceptor unit and enhance the ICT of the fluorophore. The increased bathochromic shift of **101** by the addition of a phenyl ring can be explained by the planarization of the fluorophore structure [9].

### 3.3. Aza-BODIPY Units as Linker and Acceptor Groups

Another class of compounds used for synthesizing D-π-A fluorophores is boron dipyromethene (BODIPY) dyes. The conjugated and photochemically stable core of BODIPY dyes can be used as the scaffold for D-π-A fluorophores, and several donor and acceptor groups can be attached to them [67,68,69]. Also, the BODIPY structure can be used as an acceptor unit due to the electron-withdrawing ability of BF_2_ group. The general structure and numbering of the BODIPY unit is shown in Figure 8.

There have been a number of studies that have reported the BODIPY unit containing D-π-A fluorophores. As an example, Ulrich et al. reported the synthesis of seven different types of BODIPY dye with the tetracyanobuta-1,3-diene (TCBD) acceptor, and anisole and dibutylamino donor groups [68]. The absorbance of these dyes was in the UV-vis region, between 475 and 602 nm, and they showed emission maxima between 561 and 616 nm in chloroform. The modification of C8 (meso) carbon with nitrogen results in aza-dipyrromethene boron difluoride (aza-BODIPY) dyes which show red shifted optical properties [70]. Jiao et al. reported the synthesis of aza-BODIPY fluorophores with a methoxy donor and different acceptor units that emit in the NIR region [71]. The synthesis of fluorophores **102**–**109** is presented in Figure 23. The first step is aldol condensation followed by the Michael addition of nitromethane, and finally condensation with nitromethane to yield the desired fluorophores. Fluorophore **109** was synthesized from **108** and methyliodide. The optical properties of dyes were studied in toluene, THF, chloroform, methanol and acetonitrile, and the fluorophores showed absorbance between 685 and 720 nm. The absorbances of fluorophores **107**–**109** were larger than 700 nm, and fluorophore **107** with the cyano acceptor group showed the highest bathochromic shift in toluene
λ_max_ = 720 nm. The bathochromic shift increased for fluorophores **107** and **109** with
λ_em_ = 744 nm and 756 nm in THF. This increase shows the effect of adding CF_3_ and CN acceptor groups, promoting the ICT and polarization of the molecule. The fluorophore **109** showed emission at 743 nm in methanol, at 744 nm in acetonitrile and 752 nm in chloroform [71].

In another study, Bai et al. reported aza-BODIPY fluorophores in the NIR-II region that were modified with four donor units at 1, 3, 5 and 7 positions [72]. Since the aza-BODIPY structure is a good electron-withdrawing group, it was used as the acceptor moiety in these D-π-A fluorophores and the donor units were 4-(*N*,*N*-dimethylamino)phenyl **111**, 1-ethyl-1,2,3,4-tetrahydroquinolinyl **112** and 4-julolidinyl **113**. The fluorophores were synthesized according to the similar procedure reported by Jiao et al., and the synthesis is shown in Figure 24. The absorbance maxima of fluorophores were reported as 651 nm and 799 nm for **111**, 668 nm and 830 nm for **112**, and 672 nm and 912 nm for **113**. The emission maxima of fluorophores **111**–**113** were reported as 960 nm, 1030 nm and 1060 nm, respectively, in PBS. The emission maxima were further red shifted in DMSO as 989 nm, 1036 nm and 1070 nm, respectively. The Stokes shifts of these fluorophores were also large, between 120 and 187 nm, and the quantum yields were reported as 0.16%, 0.2% and 1.0% in aqueous solutions.

### 3.4. Phenothiazine Donor

The phenothiazine molecule is used as a donor unit for D-π-A fluorophores due to the electron-donating ability of the sulfur atom, creating a stable cation upon ionization [73,74]. One of the phenothiazine containing dyes, methylene blue (MB), is an FDA-approved drug that is used for bioimaging [75,76]. Due to its conjugated structure, the phenothiazine donor containing D-π-A fluorophores have attracted attention [77,78]. Chaudhary et al. reported the synthesis of fluorophores with alkyl groups introduced at N-10 phenothiazine donor and several acceptor groups at carbons 3,7 positions, such as cyano, fluorine, acetyl and thiazole [79].

As another example, Hsieh et al. reported the synthesis of two phenothiazine donor containing fluorophores with aniline donor and nitrobenzene acceptor units [80]. The synthesis of nitrobenzene phenothiazine fluorophore is given in Figure 25. In the first step, 4-iodo nitrobenzene (**114**) was reacted with bis(pinacolato)diboron and a Pd (PPh)_3_Cl_2_ catalyst under a nitrogen atmosphere to yield boronated nitrobenzene **115**. Phenothiazine was brominated by the dropwise addition of NBS in THF to yield 3,7-dibromo phenothiazine **117**. Finally, **115** and **117** were reacted via the Suzuki coupling reaction at 105 °C for 2 days to form 3,7-Bis(4-nitrophenyl) phenothiazine **120**. The aniline phenothiazine **119** was synthesized starting from 4-bromoaniline, pinacolborane and a Pd (PPh)_3_Cl_2_ catalyst. Then, similarly, **119** was reacted with 3,7-dibromo phenothiazine (**117**) and a Pd catalyst to yield 3,7-Bis(4-aminophenyl) phenothiazine **121**. Fluorophores **120** and **121** showed emissions in the UV region in DMSO. Additionally, their emissions were also measured in DMSO with NaOH, KOH and tBuOK, which resulted in the deprotonation of the nitrogen at position 10 (N-10). When the dyes were deprotonated in DMSO under basic conditions, significant bathochromic shifts were seen. The
λ_max_ of fluorophore **120** shifted from 465 nm to 825 nm, and the negative charge on phenothiazine was stabilized by the nitrobenzene groups. Fluorophore **121** has two peaks, 350 nm and 535 nm, which shifted to 535 nm and 740 nm upon the deprotonation of N-10 [80].

Gong et al. reported the synthesis of fluorophores with the phenothiazine donor and different acceptor groups emitting in the NIR region [81]. The butterfly shape of phenothiazine promotes the solid state emission of fluorophores due to steric hindrance. The synthesis of fluorophores with a phenothiazine donor and different acceptor groups **124**–**127** is given in Figure 26. First, the alkylation of N-10 was conducted, and then **122** was reacted with phosphorous oxychloride to yield aldehyde compound **123**. In the last step, compound **123** was reacted with the four different acceptor groups in methanol under basic conditions to form fluorophores **124**–**127**. Fluorophores **124**–**127** showed absorption between 392 and 495 nm in DCM. The solid state of dyes showed emissions in the NIR region between 551 and 727 nm. The dyes **126** and **127** specifically showed bathochromic shifts due to their strong acceptor groups, their
λ_max_ were 717 nm and 727 nm, and quantum yields were 2% and 5%, respectively. The dyes **124** and **125** had the highest quantum yields of 23% and 38%, respectively, however their emissions were the most hypochromic shifted ones [81].

### 3.5. Chloroacrylic Acid Acceptor

As discussed in the previous examples, the cyanoacetic acid and carboxylic acid groups are commonly used acceptor groups in D-π-A fluorophores. In addition to these groups, chloroacrylic acid can be used as an acceptor group since it contains carboxylic acid and chloride. Our group reported the synthesis of D-π-A fluorophores with a chloroacrylic acid acceptor unit and several different donor units as indole, benzothiazole, benzo[e]indole, 2-quinoline and 4-quinoline [15]. The acceptor and donor units in these fluorophores are connected with a bisaldehyde **129** which is synthesized via Vilsmeier Haack formylation and referred to as a Vilsmeier linker. The hepta cyanine dyes containing the Vilsmeier linker between two heterocyclic units have absorbance maxima in the NIR region [82,83]. This similar scaffold was modified to synthesize D-π-A fluorophores in the NIR region, as shown in Figure 27. The fluorophores **131**–**137** showed absorption maxima in the NIR region of 790 nm, 800 nm, 815 nm, 860 nm and 970 nm, respectively. The highest quantum yield was reported for fluorophore **132** with a benzothiazole donor unit of 19%. 2-quinoline and 4-quinoline containing fluorophores **136** and **137** were significantly red shifted, which was attributed to the planarity and enhanced conjugation of the fluorophores. Emission maxima of the fluorophores **131**–**134** were reported as 810 nm, 812 nm, 824 nm and 870 nm, respectively. The emission maxima of fluorophore 135 was out of the range of the instrument. According to the TD-DFT studies, the chlorine atom at the meso position of the linker unit also contributes to the acceptor unit due to its electron-withdrawing ability. This property forms a stronger acceptor unit.

## 4. Optical Properties of Selected Compounds

### 4.1. Two Photon Absorption

In one photon absorption process, when a molecule absorbs a photon, the electrons at the ground state go into an excited state and relax back by emitting light. In the case of two-photon absorption (2PA), the molecule absorbs two photons instead of one. In this process, instead of using a high-energy photon, two lower energy photons can be used for exciting the molecule while creating an intermediate state [84]. 2PA was first introduced by M. Goppert-Mayer in 1931, who proposed that an intermediate state forms upon the absorption of a photon, and this is followed by the absorption of a second photon and moving to a higher state. This process is shown in Figure 9 by an electronic state diagram. However, due to the lack of a coherent light source, this theory could not be tested until the laser was invented in 1961 [85].

Multiphoton excitation (MPE) is a process where the compound absorbs more than two photons, however, this process is much more complicated than 2PA since the absorption of several photons may ionize the molecule, change the refractive index of the medium, and have several other effects [85]. Note that the size of the molecule should be greater than the 2PA cross-section area (δ_2PA_), which is the result of the diffraction limit such that 2PA becomes possible [8]. Fluorophores **21**–**29** were designed as two-photon-absorbing molecules due to their large surface areas, and their 2PA properties were measured in DCM with 1-3 mM solutions that have an absorption between 740 and 800 nm, which is approximately twice that of the one-photon absorption studies. The 2PA cross-section areas of fluorophores **21**–**25** were between 158 and 427 GM. The fluorophores with thiophene **22** and **24** showed two peaks different from the others; one of the peaks was at 940 nm [32]. Among the fluorophores **26**–**29**, compound **28** with cross section dicyano group showed the highest 2PA cross-section. The 2PA studies of fluorophore **29** were performed in THF with 10^−4^ M solutions. The dicyanovinyl acceptor containing dye **27** had a 2PA cross-section area of 667 GM [34]. Another example reported by Huang et al. is dicyanostilbene derivatives containing a carbazole donor unit that is used as a two-photon fluorescence temperature probe [86].

### 4.2. Aggregation-Induced Emission

Most of the luminescent molecules tend to aggregate with increased concentration in solution, resulting in decreasing fluorescence, which is defined as aggregation-caused quenching (ACQ). An example of this is fluorescein which is soluble in water but insoluble in most organic solvents. In a water–acetone mixture, as the volume of acetone is increased, the emission of fluorescein decreases as an example of quenching fluorescence by aggregation. On the other hand, aggregation-induced emission (AIE) is when the luminescent molecules come together as aggregates and emit light. In AIE, the fluorophores are not emissive when they are dissolved in solution, but they become emissive when the concentration increases, and they become aggregates. Most of the fluorophores that are emissive in solution have a mostly planar structure and they are quenched by aggregation, while AIE fluorophores are non-planar and have twisted structures [87]. AIE fluorophores are used in both photovoltaic applications and in bioimaging studies. Fluorophores with AIE properties can be designed strategically to accumulate at targeted organelles. One of the examples by Zhuang et al. reported a strategical design of fluorophores with a cationic pyridinium unit to target mitochondria for reactive oxygen species (ROS) generation [88]. Based on the positional isomerism strategy employed by the researchers, it was reported that cross section cyano group on the fluorophores with AIE induces the ROS generation. In another study, Li et al. reported that the coumarin-based fluorophores with AIE properties for H_2_S sensing [89] were not limited to the AIE, but we discuss more applications in the following section.

## 5. Applications

D-π-A fluorophores find applications in several areas such as organic solar cells and light-emitting diodes, bioimaging, biosensors and anti-cancer activity. These topics will be discussed briefly in this section.

### 5.1. Organic Solar Cells

Solar cells are clean energy sources that convert light energy to electricity. Organic solar cells have several advantages over inorganic silicon-based solar cells such as low cost, flexibility and non-toxicity [90]. Although the polymer- and fullerene-based solar cells are the most studied ones, during the last few years there has been an ongoing interest in small molecule-based organic photovoltaics [91]. The most challenging part of photovoltaic studies is obtaining a high-power conversion efficiency (PCE) which depends on the open circuit voltage (V_oc_), circuit current density (J_sc_) and fill factor (FF). The working mechanism of the solar cells has been investigated in detail over the years. The donor unit absorbs light and an exciton is generated which is followed by its diffusion and dissociation, and collection by the acceptor [92]. Dye-sensitized solar cells (DSSC) are non-toxic sources that produce clean energy [93]. The structure consists of a working electrode, counter electrode and an electrolyte. The working electrode is usually the conducting indium tin oxide (ITO) that has a coating of a semiconductor material such as TiO_2_, ZnO or SnO_2_. The organic dye is coated on the semiconducting material and absorbs the incoming light. The organic dyes can be coated on the semiconductor via different mechanisms such as covalent bonding, hydrogen bonding, van der Waals forces, electrostatic attraction and adsorption [93,94]. The interaction of dyes with the metal surface is via their anchoring groups, and the orientation of the dyes on the semiconductor surface affects the efficiency [95]. Some of the anchoring groups used in the literature are pyridyl, carboxylic acid, phosphonic acid, tetracyanate and perylene groups [93]. D-π-A fluorophores are preferred in DSSCs since they have efficient intramolecular charge transfer properties [57]. The fluorophores **83a**,**b** and **86a**,**b** reported by Michinobu et al. with a tetracyano acceptor group were used for dye-sensitized solar cells (DSSC). The TiO_2_ powder was reported to show color change when immersed in the dye solutions, therefore, the fluorophores show a surface adsorption. The device sensitized using a triphenylamine group (**86a**) showed a higher current density (0.65 mA cm^−2^) than that of **83a** (0.12 mA cm^−2^). In the case of dyes with a tetracyanoquinodimethane (TCNQ) acceptor unit, **86b** showed a current density of 1.71 mA cm^−1^, while **83b** showed a current density of 1.41 mA cm^−2^. TCNQ containing fluorophores showed better open circuit voltage than tetracyano ethylene (TCNE) containing fluorophores, and the best photon conversion efficiency was reported for fluorophore **86b**. The power conversion efficiencies of devices with fluorophores **83a** and **83b** was as low as 0.58% and 6.3%. On the other hand, the power conversion efficiencies of fluorophores **86a** and **86b** were 20% and 25%.

Fluorophores **90a** and **90b** (Figure 19), which have a quinoxaline linker and indoline acceptors, were discussed in the synthesis part. The photovoltaic properties of solar cells with dyes were studied on a double layer TiO_2_ film as the anode and Pt as the counter-electrode (CE). The absorption maxima of dyes on the TiO_2_ film were reported as 502 nm and 508 nm, respectively. For the iodide-based electrode, the V_oc_ values were similar, while the power conversion efficiency of **90b** was higher than **90a**. In the case of a cobalt-based electrode, the V_oc_ and power conversion efficiencies were higher for **90b** with a bulkier donor unit.

The photovoltaic performances of the A-D-A-type three fluorophores **93**–**95** were investigated using bulk heterojunction cells where PEDOT: PSS was used as a hole transport layer and [6,6]-phenylC_71_ butyric acid methyl ester (PC_71_BM) were used as an acceptor. The devices had bulk heterojunction architecture with ITO and calcium electrodes. While **93** had low solubility, the other two fluorophores showed better solubilities. The PCE of the fluorophores were found to be between 0.63 and 1.36%, with **95** being the highest [62].

### 5.2. Bioimaging

Previously discussed fluorophores **63** and **64** with carbazole donor and pyrylium and quinolinium acceptors, and fluorophores **68** and **69** with diphenylamine donor and pyrylium and quinolinium acceptors, were reported to show AIE properties. The optical properties of fluorophores were studied in DMSO/toluene mixtures, and, upon the increase in toluene fractions, the dyes aggregated and the fluorescence intensity got higher. One of the examples of using AIE fluorophores for bioimaging is using them as mitochondria-targeting agents. Mitochondria, which are the power cells of the human body, are negatively charged in the inner membrane which is close to matrix [87]. Therefore, positively charged and lipophilic fluorophores are used as targeting agents [96,97]. Fluorophores **63**, **64**, **68** and **69** with positively charged acceptor groups showed specificity to mitochondria and the images of HeLa cells stained with four different fluorophores are presented in Figure 10.

The AIE properties of previously discussed fluorophores **120** and **121** were studied by Hsieh et al. in THF/water solvent with different ratios. The dye **121** showed AIE upon acidic conditions due to the protonation of nitrogen. The phototoxicity experiments were conducted with HeLa cells and the singlet oxygen acceptor diphenylisobenzofuran (DPBF). While dye **120** did not show any phototoxicity effect, dye **87** was effective. However, since the absorbance of **121** was in the UV region, the indicators DPBF and TEMPO, which are photoresponsive agents, may also be excited [80].

Fluorophores in NIR-I and NIR-II regions are preferred for bioimaging studies since there is less autofluorescence from biomolecules in these regions and penetration depth is higher [5]. The NIR-II window between 1000 and 1700 nm is reported to have improved imaging quality over the NIR-I window [98]. So far, several D-π-A fluorophores have been synthesized in the NIR region with different scaffolds [99,100]. Aza-BODIPY fluorophores **111**–**113** in the NIR-II region that were reported by Bai et al. were discussed previously [72]. The fluorophores contain three different donor units, namely 4-(*N*,*N*-dimethylamino) phenyl **111**, 1-ethyl-1,2,3,4-tetrahydroquinolinyl **112** and 4-julolidinyl **113**. Fluorophore **113** was selected for bioimaging studies, as presented in Figure 11, since it has the highest red shift. However, it showed low water solubility, therefore, the fluorophore was prepared as a nanoparticle (NP) by encapsulation into the Pluronic F-127 matrix. The NPs showed absorption maxima at 858 nm and emission maxima at 1062 nm. The imaging depth was reported as 8 mm. The imaging studies were performed according to FDA-approved indocyanine green (ICG) dye and IR-1061 dye. It was reported that fluorophore **113** NPs showed high imaging ability. After injection over 6–8 h, the tumor signal was increased. The ex vivo biodistribution studies showed that the dye accumulates in the liver and spleen, in addition to in the tumor tissue.

Compounds with a quinolinium acceptor unit were reported by Zheng et al. as a probe to detect viscosity in cirrhotic liver tissue [101]. Due to the flexible polymethine chain in between, the dimethyl phenyl group can rotate, which is known as a twisted intramolecular charge transfer, and the molecule loses some of its absorbed energy by this rotation in low-viscosity solvents.

Cesaretti et al. reported the same scaffold as RNA selective donor acceptor styryl dyes with positively charged quinolinium and pyridinium acceptor units **138** and **139** [102]. They performed spectrophotometric and fluorimetric titrations with calf thymus DNA and tRNA to study the interaction between the dye and nucleic acids in ETN (1 mM EDTA, 10 mM Tris-HCl, 10 mM NaCl) buffer solution. While the absorbance intensity of dye containing a pyridinium acceptor decreases, a quinolinium acceptor unit containing dye showed an increase. The in vitro studies of two dyes were performed with A549 cells using fluorescence microscopy with a 10 uM concentration. The dyes were specifically found at the perinuclear portion of the cytoplasm, and punctuated structures within the nuclei were lit up, as shown in Figure 12.

### 5.3. Organic Light Emitting Diodes

Organic light-emitting diodes (OLEDs) are LEDs with organic compounds emitting light upon electric current [103]. The devices consist of an anode which is commonly ITO, a cathode and an organic layer of molecules placed between those layers. OLEDs have found applications in several areas such as lightning, OLED TVs, cell phones and digital cameras [104]. One of the advantages of using organic molecules for LED devices is that inorganic compounds themselves or their purification process may be harmful to the environment [105]. OLEDs have advantages that they are non-toxic, and they do not face the problem of containing metals that may become extinct. The pioneering work on OLEDs was reported by Tang et al., in which the electroluminescent behavior of double layer diamine, Alq^3^, was investigated using a Mg:Ag alloy cathode [106]. Although polymers were mainly used as electroluminescent organics, small organic molecules have also attracted attention over the last few years due to their low cost, low turn-on voltage and ease of modification of the structure. The dyes that show AIE properties due to the restriction of the intramolecular rotation of bonds find application in OLEDs [107,108].

Carbazoles are one of the organic compounds used for OLEDs starting with their usage as polymers [109,110]. Previously discussed star-shaped fluorophores **79a**–**c** with carbazole units have been used in OLED applications. The monomers undergo irreversible oxidation and, while **79a** and **79b** showed similar characteristics on the CV, **79c** shifted to lower potential. Optical spectra of the polymers were investigated for the p doping process. All polymers were reported as neutral at a potential of ca. −4.0 V, showing an absorption peak for **79a** due to
π-π* transition. Optical band gaps of the polymers prepared from D-
π-A fluorophores were reported as 3.05 eV, 3.12 eV and 2.55 eV for **79a**–**c**. The lower band gap of **79c** is due to the thienyl group promoting a higher conjugation.

### 5.4. Anticancer Activity

Fluorophores **131**–**134** with a chloroacrylic acid acceptor and different donor units, as presented in Figure 13, were studied for their cytotoxicity against breast cancer cell lines HCC-1806, HCC-70 and BT-20, using an MTT assay. The cytotoxicity of the fluorophores was compared with the known anticancer drug docetaxel and the fluorophores showed higher activity than docetaxel. Especially for fluorophore **132** with a 5-bromoindole donor unit, the IC_50_ value was reported as 0.717
μM, and 0.940
μM and 1.470
μM for HCC-1806, HCC-70 and BT-20 cells, respectively, whereas docetaxel had IC_50_ values of 50.9, 35.6 and 2.41
μM. In another study reported by Bai et al., the aza-BODIPY fluorophores were studied. The fluorophores contain three different donor groups, namely 4-(*N*,*N*-dimethylamino) phenyl **111**, 1-ethyl-1,2,3,4-tetrahydroquinolinyl **112** and 4-julolidinyl **113**, as discussed before. Fluorophore **113** was selected for in vivo imaging due to its emission in the NIR-II region, and NPs were prepared in a Pluronic F-127 matrix. The fluorophore did not show any cytotoxicity compared to standard XTT analysis.

### 5.5. Biosensors

In addition to the applications discussed before, another application of D-π-A fluorophores is their usage as metal and toxic ion sensors. Among the heavy metals, some of them show serious effects on the human body in the case of high exposure, such as Cu^+2^, Hg^+2^ and Pb^+2^. Cu^+2^ is an essential divalent in the human body since it takes roles in the metabolic pathway, however, in the case of high concentration it is toxic and can cause diseases like Alzheimer’s and Parkinson’s disease [111]. Nguyen et al. reported Cu^+2^-detecting donor acceptor-type fluorophores with a picoline moiety and dicyanometylenedihydrofuran (DCDHF) acceptor [111]. While the absorbance of this dye is 400 nm without Cu^+2^, upon addition of Cu^+2^ the absorbance red shifts to 590 nm. Another D-π-A fluorophore reported for Cu^+2^ detection containing coumarin and benzothiazole showed a detection limit between 4.0 and 5.7 ppb [112]. In addition to the heavy metals, cyanide is another toxic molecule for the human body due to its high electron-withdrawing effect, and it can bind to essential metals and cause disorders [113]. Therefore, there is an ongoing interest in cyanide detection in water and in vivo using small organic D-π-A molecule probes [114,115,116,117].

Another example is hypochlorite ions that are used for bleaching and disinfecting processes which are harmful for human health, causing cancer, arthritis and tissue damage [75]. Huang et al. reported the usage of an acetylated methylene blue probe for the detection of hypochlorite ions in water [75]. The studies showed that, upon the addition of a hypochlorite ion to the carbonyl carbon, the amide bond of the probe is cleaved, releasing the methylene blue and resulting in a color change of the solution.

In another study, Arachchige et al. reported quinoline acceptor containing fluorophores as candidates for investigating NADH dynamics and redox process [27]. As the NADH concentration is increased, the florescence intensity of the probes increases. While NADH becomes oxidized, the 3-quinolinium unit is reduced to a 1,4-dihydroquinoline unit. Upon reduction of the quinolinium unit, the extended conjugation is regained, resulting in higher fluorescence signals as presented in Figure 14.

Last but not least, donor acceptor fluorophores also find an application as ratiometric probes for biosensing [118]. Ratiometric fluorophores have two or more peaks with changing intensities due to a change in environmental factors such as pH or various analytes [119]. As an example, Munan et al. reported the synthesis of fluorophores called IndiFluors which consists of a conjugated rigid D-A-D system modified with a mitochondria-targeting triphenylphosphine moiety for ratiometric pH imaging during mitophagy [120]. In addition to pH change, an example of analyte response has been reported by Yuan et al., who designed a lysosome-targeting probe containing a morpholine moiety to detect leucine aminopeptidase activity [121]. The emission at 520 nm decreases while a new peak at 580 nm increases upon treatment with leucine aminopeptidase. Table 1 presented below, highlights absorbance, emission and quantum yield of selected fluorophores discussed above in this paper.

**Table 1 biomolecules-15-00119-t001:** Optical properties of selected fluorophores. * Fluorophores with more than one absorbance wavelength maxima.

Compound Number	Solvent	λ_abs_ (nm)	λ_em_ (nm)	QY (Φ)
**1a**	Toluene	349	418	0.48
	DCM	347	449	0.016
	Acetone	344	461	0.010
**1b**	Toluene	385	448	0.24
	DCM	386	488	0.20
	Acetone	381	500	0.070
**1c**	Toluene	369	424	0.48
	DCM	368	460	0.070
	Acetone	363	488	0.010
**1d**	Toluene	379	434	0.49
	DCM	380	492	0.40
	Acetone	373	515	0.15
**1e**	Toluene	407	472	0.29
	DCM	403	528	0.23
	Acetone	397	547	0.19
**1f**	Toluene	396	454	0.37
	DCM	395	506	0.30
	Acetone	387	531	0.17
**5a**	DCM	347	420	0.51
**5c**	DCM	312	612	0.54
**5d ***	DCM	314, 394	590	0.41
**6a ***	DCM	308, 381	447	0.39
**6c ***	DCM	321, 406	647	0.07
**6d ***	DCM	306, 333, 403	630	0.37
**7a ***	DCM	278, 383	462	<0.01
**7c ***	DCM	328, 419	607	0.80
**7d ***	DCM	305, 342, 416	583	0.79
**9a**	CHCl_3_	436	491	0.002
**9b**	CHCl_3_	455	531	0.006
**9d**	CHCl_3_	534	652	0.61
**10a**	CHCl_3_	495	548	0.003
**10b**	CHCl_3_	514	579	0.006
**10d**	CHCl_3_	595	710	0.20
**14a**	DMF	413	-	-
**14b**	DMF	444	-	-
**16a**	DMF	433	-	-
**16b**	DMF	446	-	-
**18a**	DMF	459	-	-
**18b**	DMF	479	-	-
**21 ***	DMF	332, 401	577	0.55
**22**	DMF	363	587	0.58
**23**	DMF	345	630	0.37
**24**	DMF	370	617	0.38
**25**	DMF	339	551	0.67
**26**	DMF	372	448	0.64
**27**	DMF	453	549	0.29
**28**	DMF	420	533	0.69
**29**	DMF	412	516	0.49
**37a**	DMF	402	536	0.22
**37b**	DMF	415	544	0.56
**40c**	EtOH	396	622	0.06
	DMSO	402	582	0.85
	CHCl_3_	399	532	0.72
	Dioxane	391	488	0.52
	Toluene	394	473	0.86
**40d**	EtOH	476	680	0.12
	DMSO	488	710	0.19
	CHCl_3_	486	608	0.45
	Dioxane	467	583	0.30
	Toluene	474	566	0.26
**40g**	EtOH	370	563	0.71
	DMSO	373	521	0.72
	CHCl_3_	381	492	0.54
	Dioxane	364	456	0.73
	Toluene	374	450	0.60
**63**	DMSO	398	-	0.014
**64**	DMSO	448	-	0.004
**68**	DMSO	458	-	0.09
**69**	DMSO	507	-	0.03
**83a**	DCM	570	-	-
**83b**	DCM	600	-	-
**86a**	DCM	759	-	-
**86b**	DCM	750	-	-
**97**	CHCl_3_	570	788	-
**98**	CHCl_3_	509	718	-
**99**	CHCl_3_	580	790	-
**100**	DCM	485	-	-
**101**	DCM	667	-	-
**106**	Toluene	709	740	0.38
	CHCl_3_	704	741	0.34
	THF	709	744	0.28
	MeOH	700	739	0.21
**107**	Toluene	720	754	0.36
	CHCl_3_	716	755	0.29
	THF	718	756	0.22
	MeOH	710	750	0.16
**109**	Toluene	Not soluble	Not soluble	-
	CHCl_3_	712	752	0.14
	THF	714	750	0.16
	MeOH	706	743	0.14
**111 ***	PBS (20% DMSO)	651, 799	960	0.0016
**112 ***	PBS (20% DMSO)	668, 830	1030	0.002
**113 ***	PBS (20% DMSO)	672, 910	1060	0.01
**131**	EtOH	780	798	0.1
**132**	EtOH	784	804	0.08
**133**	EtOH	810	825	0.08
**134**	EtOH	800	812	0.13
**135**	EtOH	815	824	0.16
**136**	EtOH	860	870	0.17
**137**	EtOH	970	-	-
**138**	DCM	557	705	0.25
	EtOH	502	709	0.049
	MeOH	492	712	0.027
**139**	DCM	591	679	0.053
	EtOH	548	678	0.016
	MeOH	542	686	0.006

## 6. Conclusions

In this review, the synthesis, properties and applications of D-π-A fluorophores with various donor and acceptor units were discussed. The main objective was to give examples of the synthesis and optical properties of D-π-A fluorophores since 2014, and to understand the effect of the donor and acceptor groups on the ICT. It was mentioned earlier that donor and acceptor groups, and the
π linker, have significant effects on ICT and the geometry of the molecule. Although it was reported that the planarization enhances ICT and promotes the red shift, there are some examples where the rotation of one of the units promotes ICT, showing the importance of the relation between the donor and acceptor units through ICT. The most used donor groups were diphenylamine, dimethylamine and alkoxy, while the acceptor groups of dicyano vinyl, nitro and pyridine were used. The careful modification of donor and acceptors groups makes these fluorophores useful in several areas such as solar cells, OLEDs, two-photon absorption, aggregation-induced emission, cytotoxicity studies, in vivo imaging and biosensors. In conclusion, the D-π-A fluorophores with highly conjugated structures and strong donor and acceptor groups are promising molecules in the bioanalytical and biomedical fields.

## Figures and Tables

**Figure 1 biomolecules-15-00119-f001:**
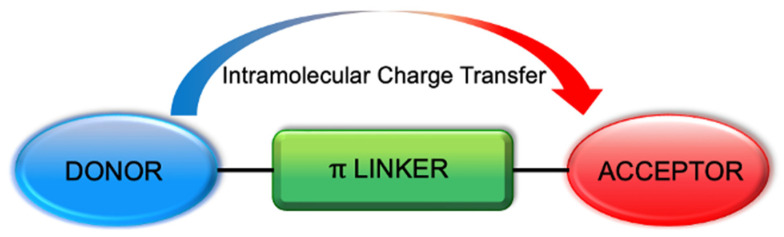
Representation of D-π-A fluorophores.

**Scheme 1 biomolecules-15-00119-sch001:**
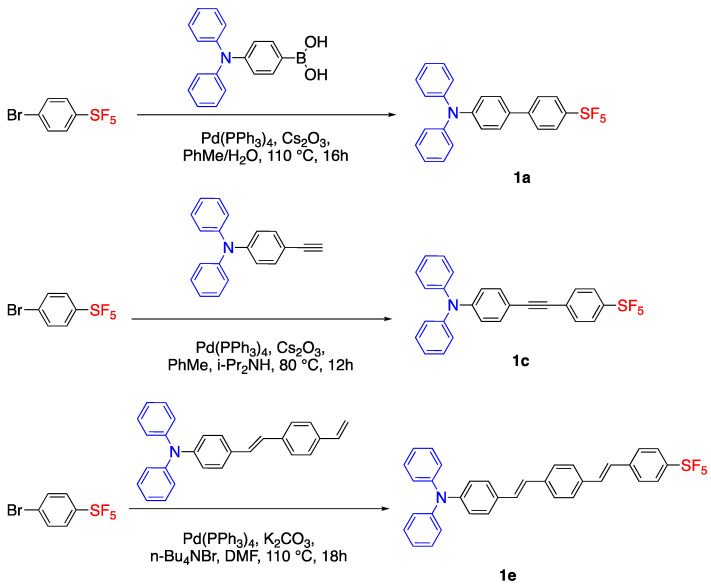
Synthesis of dyes **1a**, **1c** and **1f** via Suzuki and Heck coupling reactions.

**Figure 2 biomolecules-15-00119-f002:**
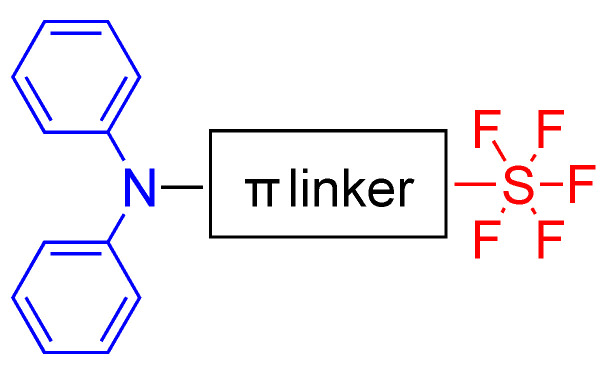
Representation of the D-π-A system containing SF_5_.

**Scheme 2 biomolecules-15-00119-sch002:**
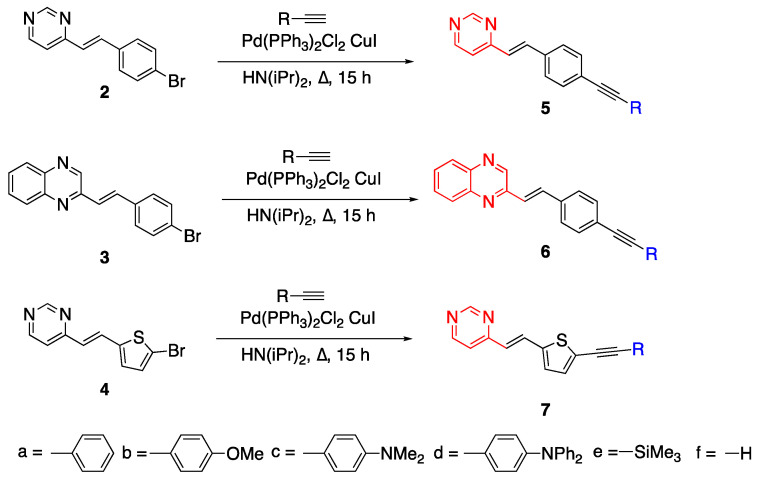
Synthesis of 2iazine c2ntaining fluorophores **5**–**7**.

**Figure 3 biomolecules-15-00119-f003:**
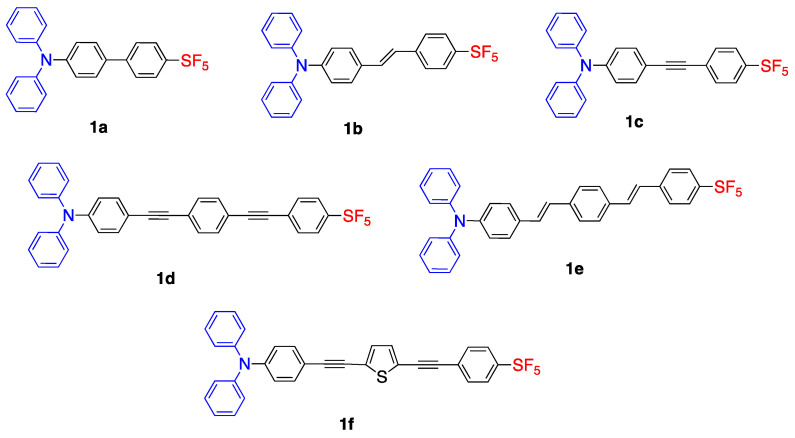
D-π-A dyes’ diphenylamino donor, SF_5_ acceptor and different π linkers.

**Figure 4 biomolecules-15-00119-f004:**
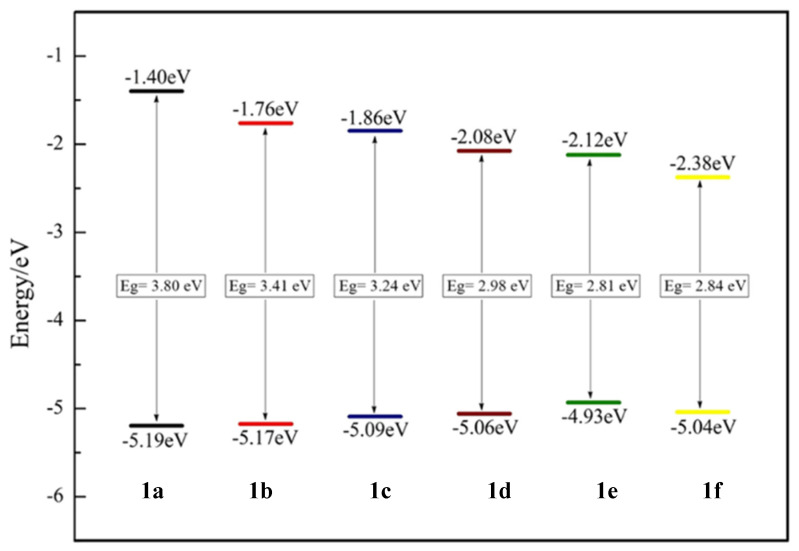
Energy band gap of fluorophores with SF_5_ acceptor from **1a**–**f** [9].

**Scheme 5 biomolecules-15-00119-sch005:**
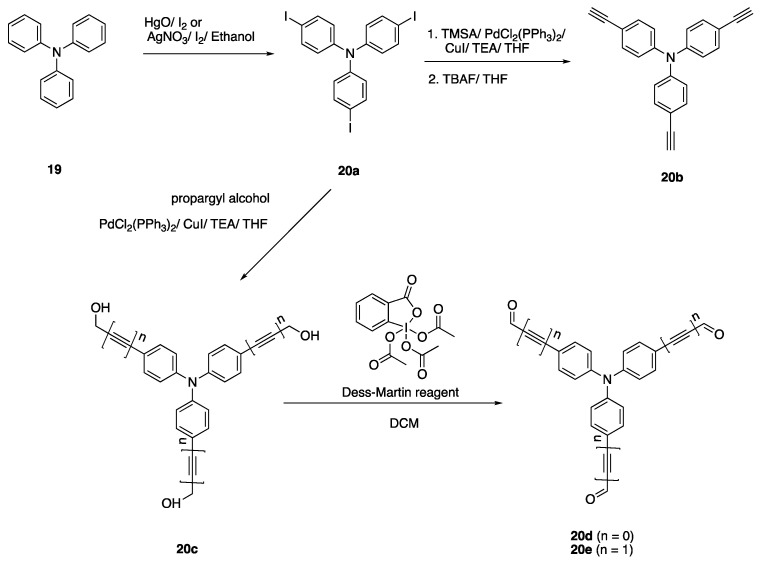
Synthesis of triphenylamine core structures.

**Figure 6 biomolecules-15-00119-f006:**
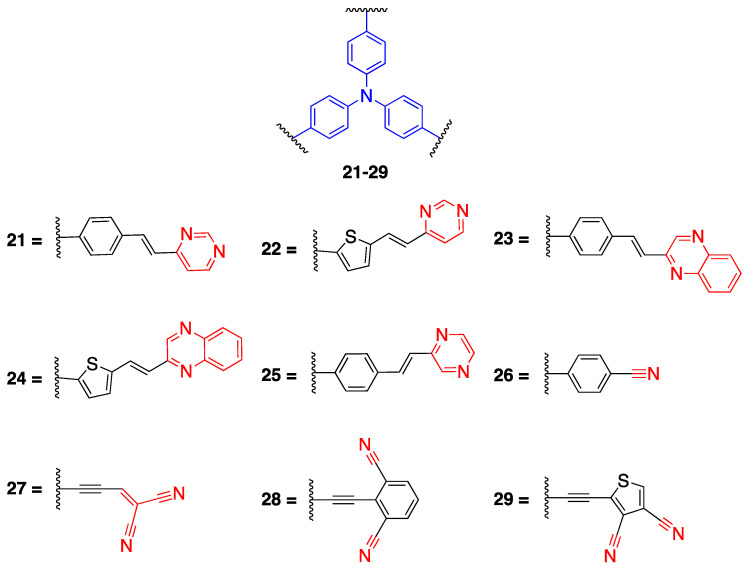
Structures of fluorophores with triphenylamine core.

**Scheme 6 biomolecules-15-00119-sch006:**
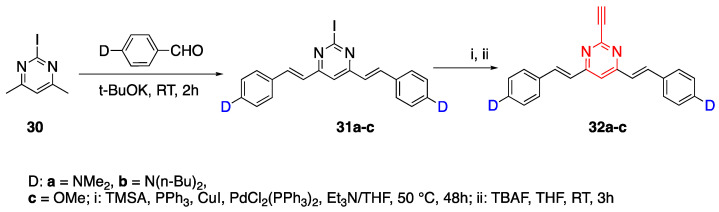
Synthesis of fluorophores with pyrimidine acceptor core.

**Scheme 7 biomolecules-15-00119-sch007:**
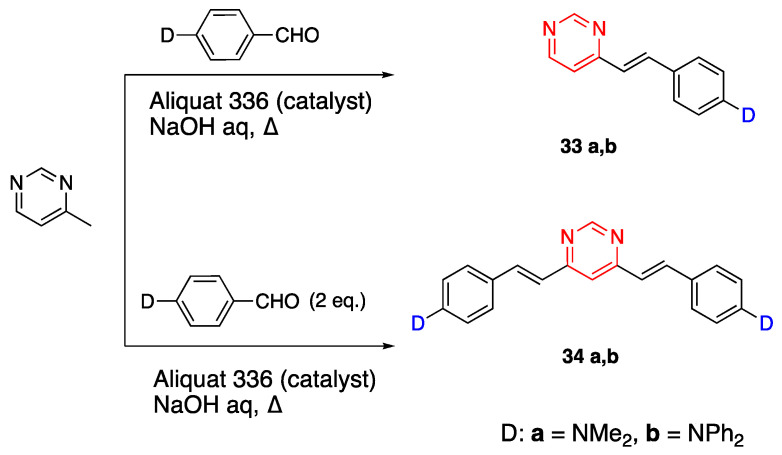
Synthesis of fluorophores with pyrimidine core as acceptor unit.

**Figure 7 biomolecules-15-00119-f007:**
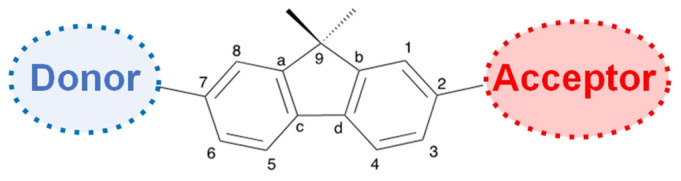
Structure of fluorene containing fluorophores.

**Scheme 8 biomolecules-15-00119-sch008:**
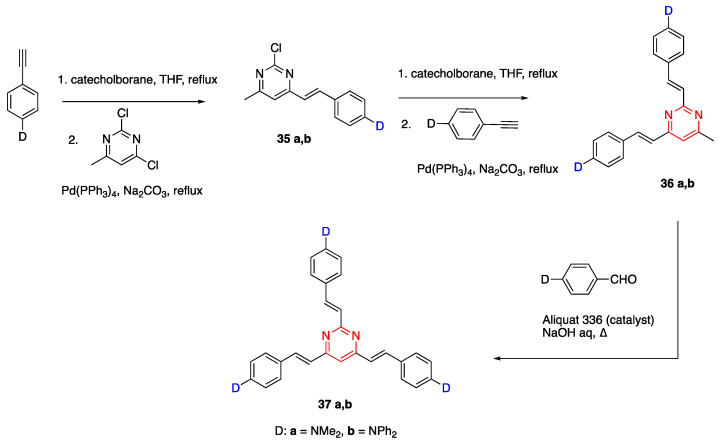
Synthesis of tripolar fluorophore with pyrimidine acceptor core.

**Figure 8 biomolecules-15-00119-f008:**
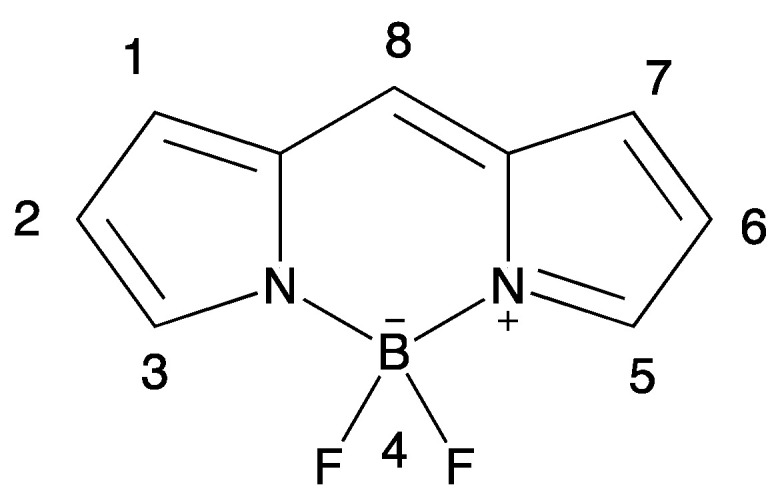
Structure of BODIPY unit.

**Scheme 9 biomolecules-15-00119-sch009:**
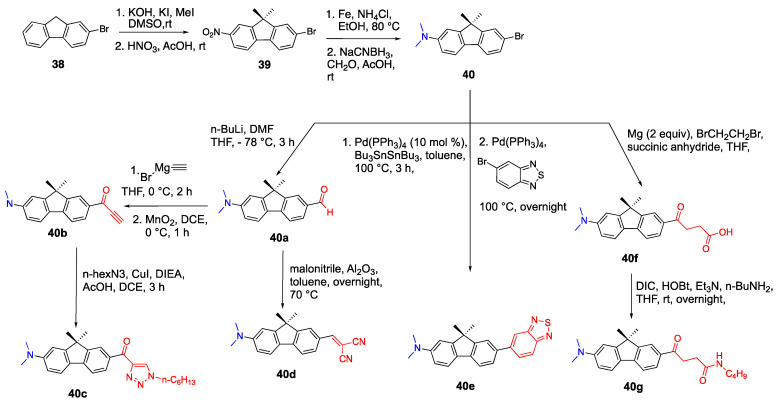
Synthesis of fluorene containing fluorophores.

**Figure 9 biomolecules-15-00119-f009:**
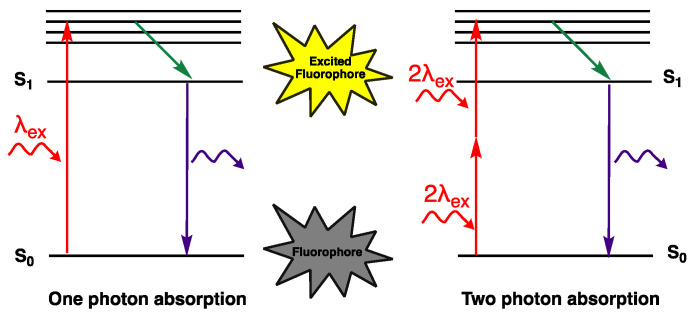
Electronic diagram showing one- and two-photon absorption processes.

**Scheme 10 biomolecules-15-00119-sch010:**
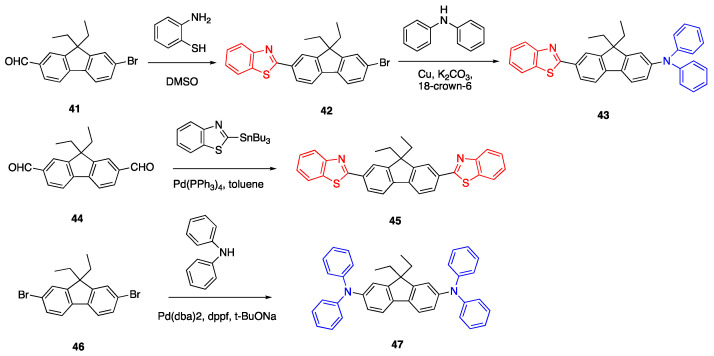
Synthesis of D-π-A fluorophores with benzothiazole and diphenylamino groups.

**Figure 10 biomolecules-15-00119-f010:**
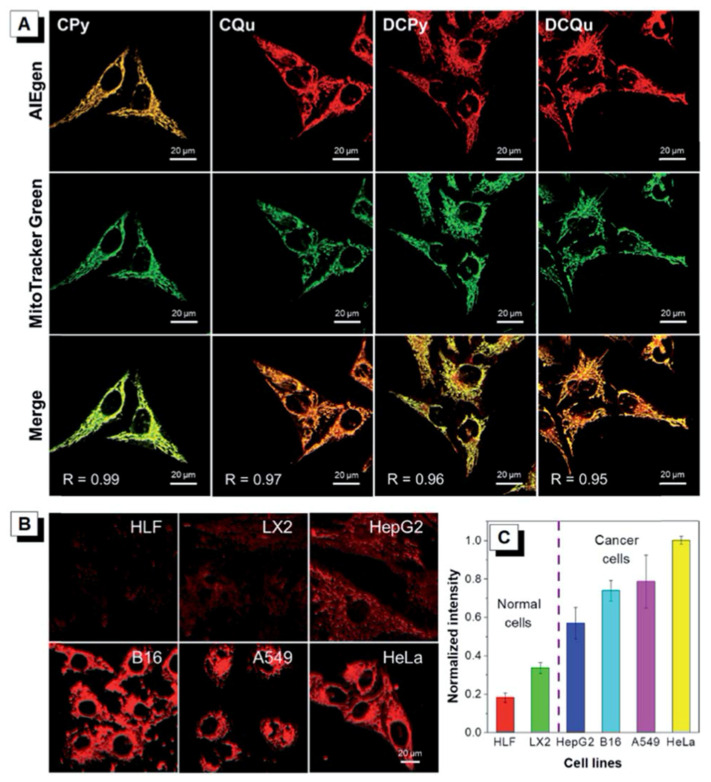
(**A**) HeLa cells stained with carbazole containing fluorophores **63**, **64**, **68** and **69** (from left to right), Mitotracker Green and merged images. (**B**) Fluorescence images of normal cells (HLF and LX2) and cancer cells (HepG2, B16, A549 and HeLa) stained with fluorophore **69** (1
 μM). (**C**) Relative fluorescence intensity of different cells incubated with fluorophore **69** for 30 min. (Reproduced from open access article [49]).

**Scheme 11 biomolecules-15-00119-sch011:**
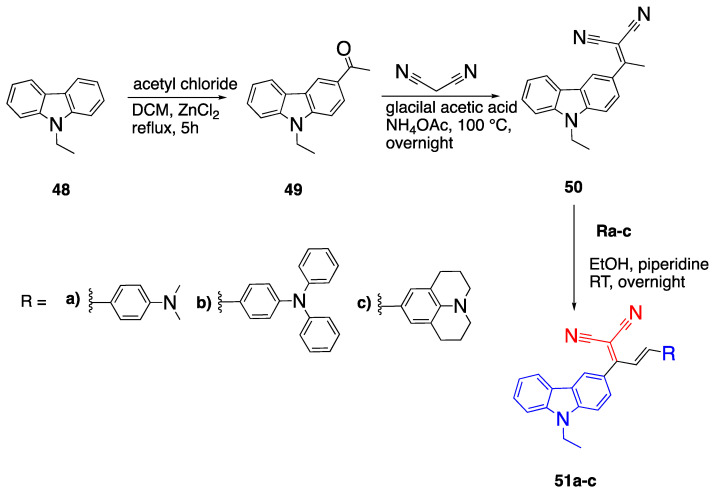
Synthesis of fluorophores **51a**–**c** with carbazole donor unit.

**Figure 11 biomolecules-15-00119-f011:**
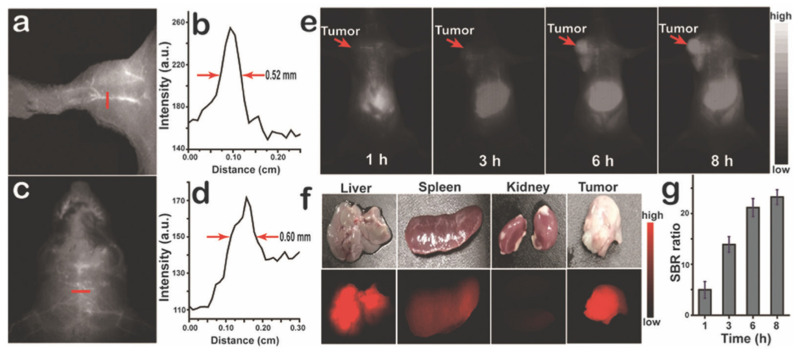
NIR-II images of (**a**) mouse hind limb (**c**) brain. Emission intensity profiles of the images (**a**,**c**), respectively, in (**b**,**d**), (**e**) NIR-II images of 4T1 tumor at different times after the injection, (**f**) ex vivo studies of fluorophore **113** NPs at 4 h under 808 nm laser radiation, (**g**) signal-to-background ratio of tumor at different times. (Reproduced with permission from [72]).

**Scheme 12 biomolecules-15-00119-sch012:**
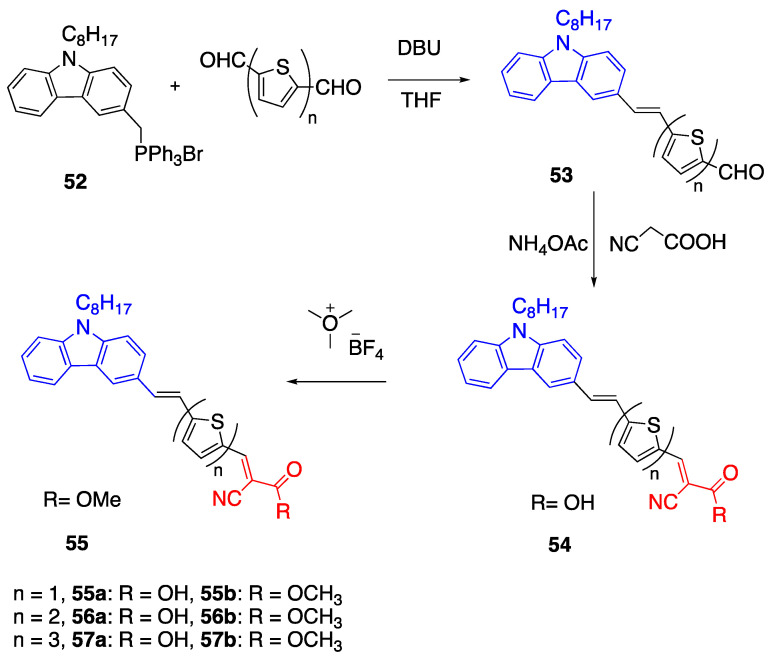
Synthesis of fluorophores **55**–**57** with alternating length of thiophene bridge.

**Figure 12 biomolecules-15-00119-f012:**
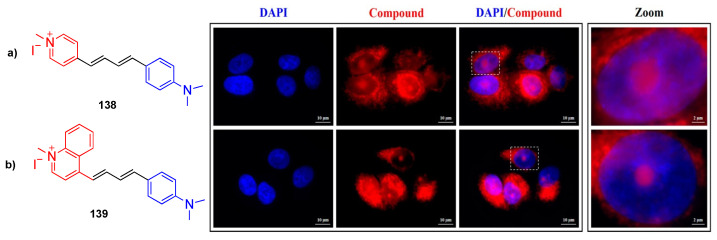
Fluorescence microscopy images of A549 cells with DAPI, (**a**) styryl dye with pyridinium acceptor and (**b**) styryl dye with quinolinium acceptor. Image reproduced from open access article [102].

**Scheme 13 biomolecules-15-00119-sch013:**
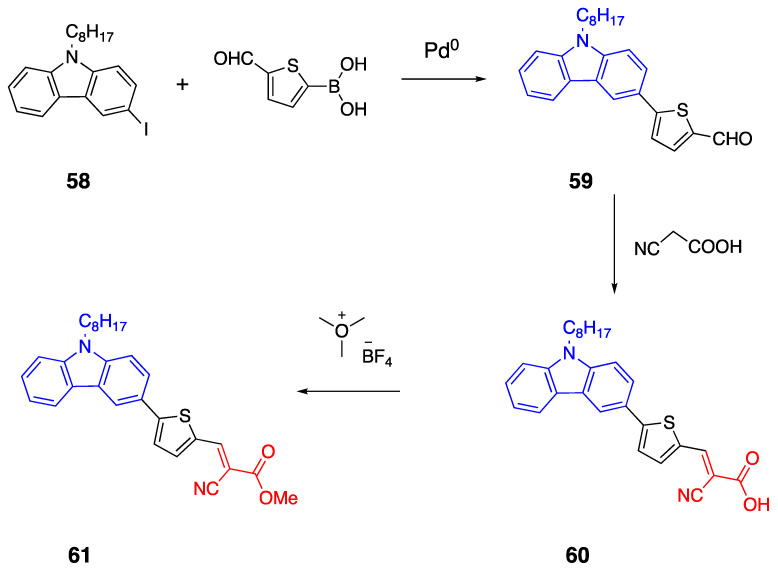
Synthesis of fluorophores **61a** and **61b** with thiophene linker.

**Figure 13 biomolecules-15-00119-f013:**
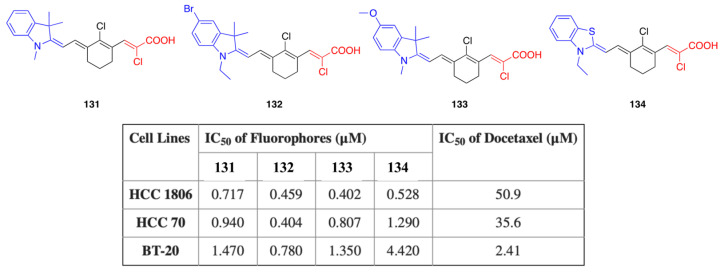
Cytotoxicity data of fluorophores **131**–**134** compared to docetaxel, against triple negative breast cancer cell lines (image reproduced from [15]).

**Scheme 14 biomolecules-15-00119-sch014:**
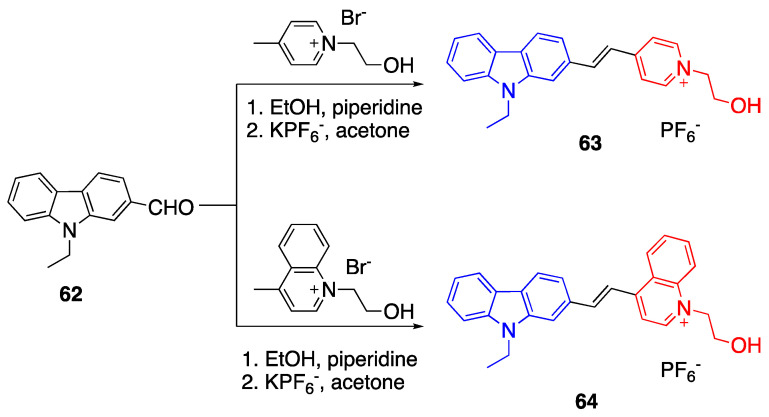
Synthesis of fluorophores **63** and **64** with carbazole donor unit.

**Figure 14 biomolecules-15-00119-f014:**
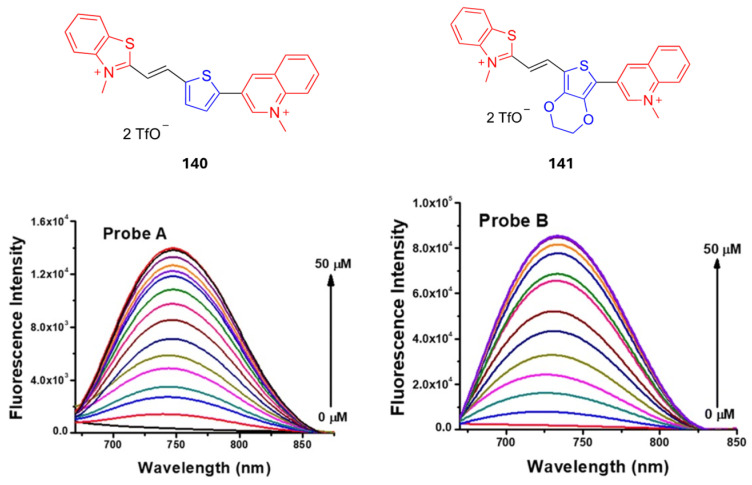
Emission profiles of probes containing quinolinium units **140** (Probe A) and **141** (Probe B), each with 10
μM, in varying NADH concentrations of 0–50 μ M in pH 7.4 phosphate buffer containing 10% DMSO. (Image reproduced with permission from [102]).

**Scheme 15 biomolecules-15-00119-sch015:**
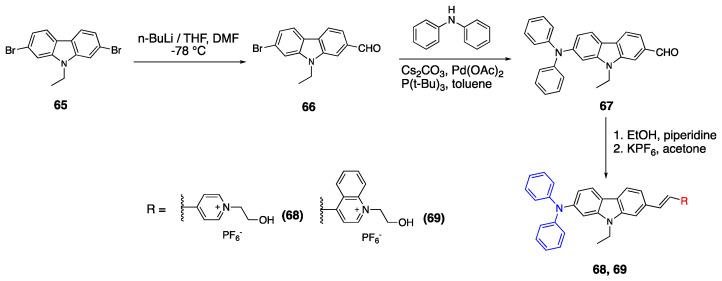
Synthesis of fluorophores **68** and **69** with carbazole linker unit.

**Scheme 16 biomolecules-15-00119-sch016:**
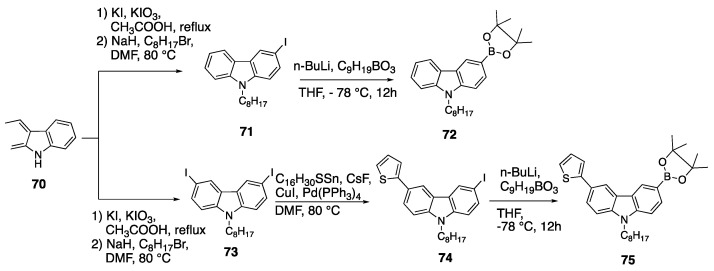
Synthesis of monoiodinated and diiodinated carbazole moieties.

**Scheme 17 biomolecules-15-00119-sch017:**
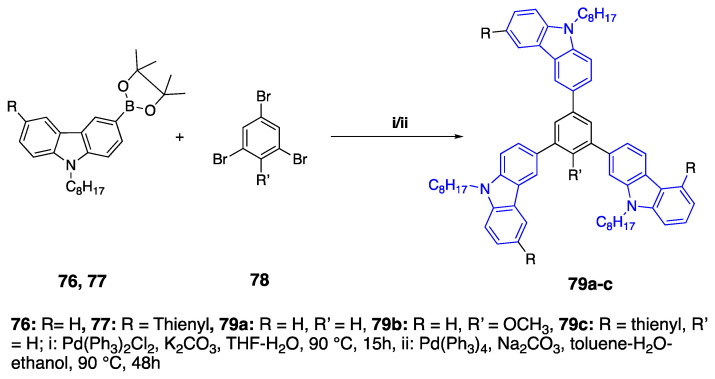
Synthesis of star-shaped carbazole donor containing fluorophores.

**Scheme 18 biomolecules-15-00119-sch018:**
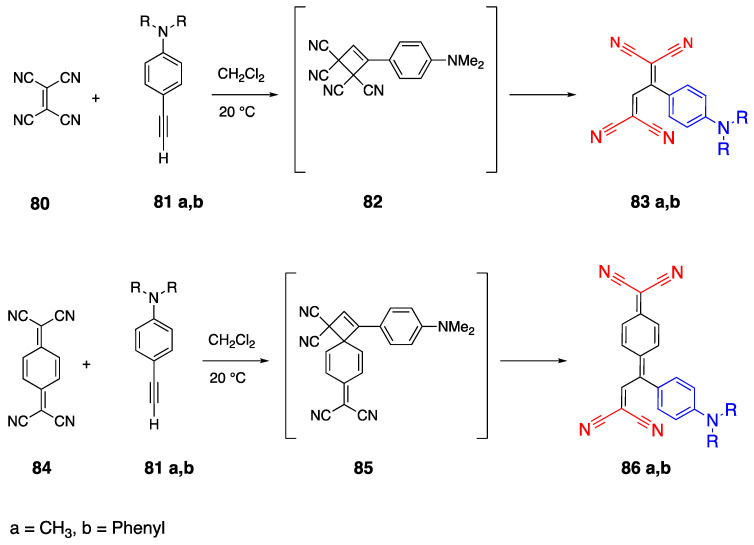
Synthesis of fluorophores **83a**,**b** and **86a**,**b** with tetracyano acceptor group.

**Scheme 19 biomolecules-15-00119-sch019:**
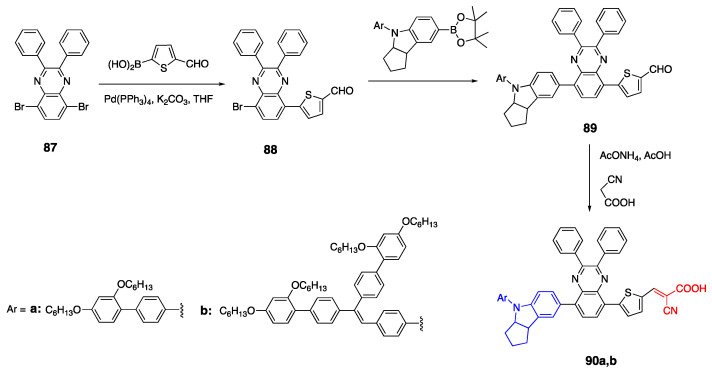
Quinoxaline linker containing fluorophores with cyanoacetic acid acceptor unit and bulky donor units.

**Scheme 20 biomolecules-15-00119-sch020:**
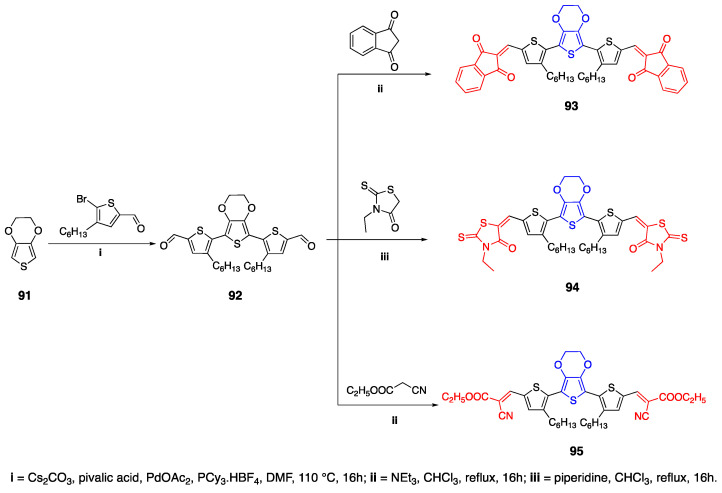
Synthesis of EDOT containing fluorophores **93**–**95**.

**Scheme 21 biomolecules-15-00119-sch021:**
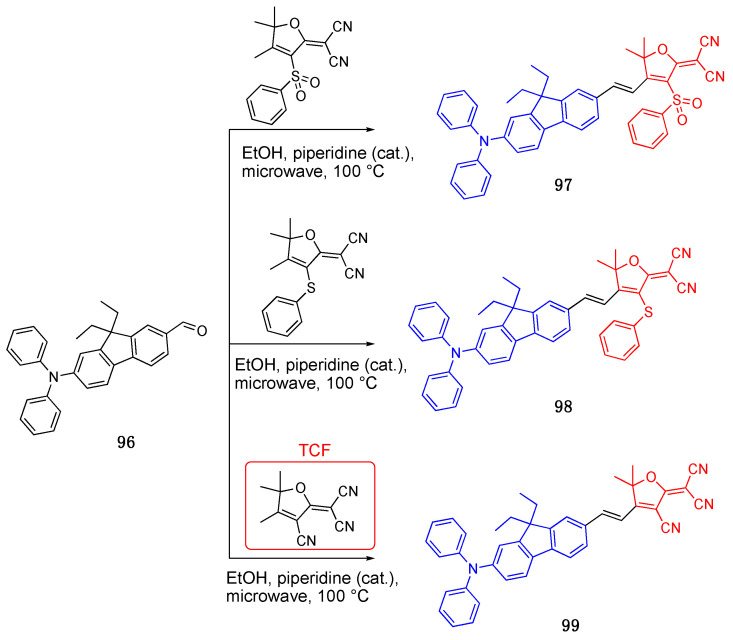
Synthesis of fluorophores **97**–**99** with TCF acceptor unit.

**Scheme 22 biomolecules-15-00119-sch022:**
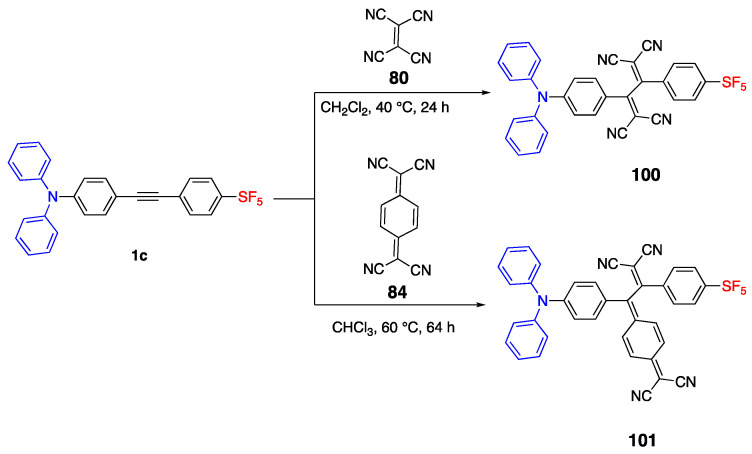
Synthesis of tetracyano moiety containing fluorophores.

**Scheme 23 biomolecules-15-00119-sch023:**
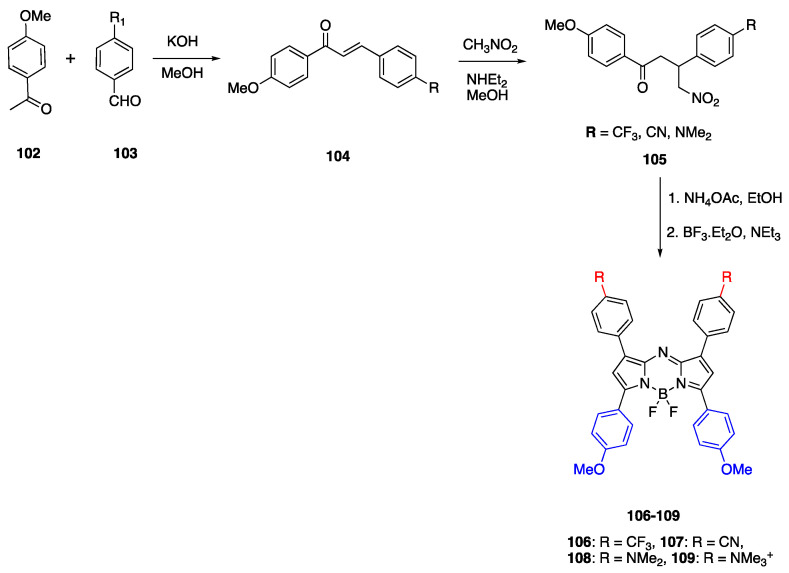
Synthesis of aza-BODIPY dyes with CF_3_, CN, NMe_2_ and NMe_3_^+^ groups **106**–**109**.

**Scheme 24 biomolecules-15-00119-sch024:**
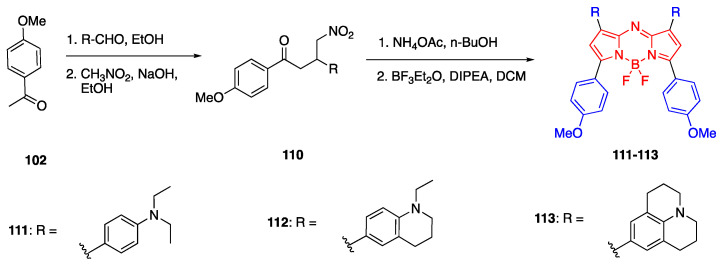
Synthesis of aza-BODIPY fluorophores in NIR-II region.

**Scheme 25 biomolecules-15-00119-sch025:**
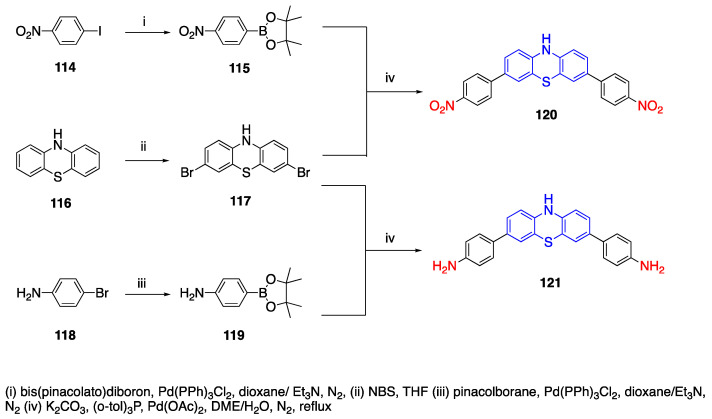
Synthesis of fluorophores with phenothiazine donor **120** and **121**.

**Scheme 26 biomolecules-15-00119-sch026:**
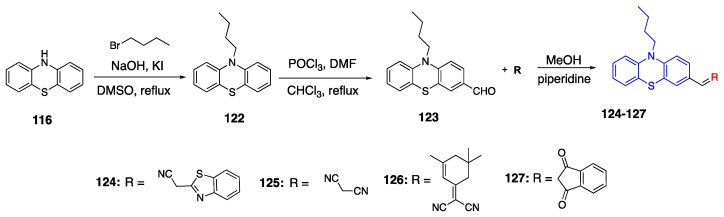
Synthesis of phenothiazine fluorophores.

**Scheme 27 biomolecules-15-00119-sch027:**
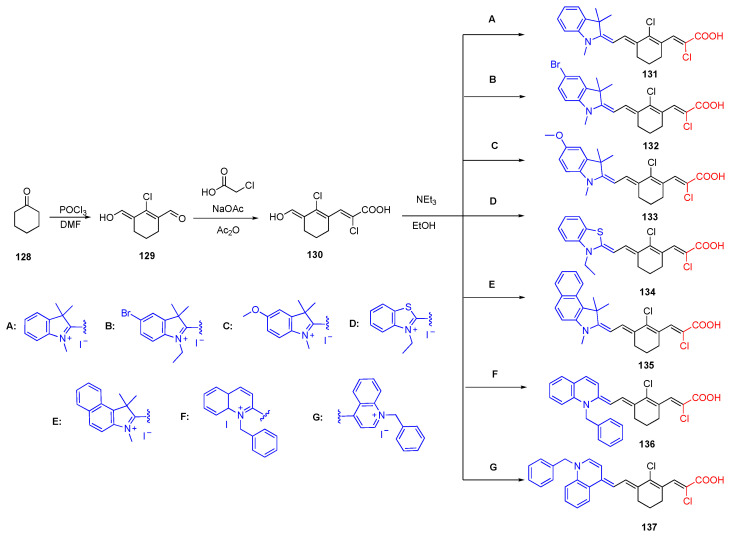
Synthesis of fluorophores **131**–**137** with chloroacrylic acid moiety.

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
