# Peer review of "Roadmap for Designing Donor- -Acceptor Fluorophores in UV-Vis and NIR Regions: Synthesis, Optical Properties and Applications"

_biomolecules, 2025, doi:10.3390/biom15010119_

Round 1
Reviewer 1 Report
Comments and Suggestions for Authors
This is an interesting and worthy publication review. However, I found a number of typos and minor errors in the text that should be corrected:
- Lines 91-95, 448-449 units (eV) should be added
- Figure 4: Cs2O3 should be replaced by Cs2CO3
- Scheme 3: cyanovinyl should be replaced by malonitrile
- Line 168: pi should be replaced by p
- Scheme 9 : transformation 39-à 40 is not clear. What is the source of “R”?
- Line 472: term “tetracyano acceptor” is not correct. I suggest something like tetracyanobutadienyl.
- - Line 666: what does the term "Vilsmeier linker" mean?
- Scheme 26: “R” as a reagent in the transformation of 123à124-127 is not correct.
- Scheme 27: What are “heterocyclic salts”?
- In some schemes the formulas are not correctly written e.g. CHCl3 instead of CHCl3
Author Response
Reviewer 1
This is an interesting and worthy publication review. However, I found a number of typos and minor errors in the text that should be corrected:
- Lines 91-95, 448-449 units (eV) should be added
- Figure 4: Cs2O3 should be replaced by Cs2CO3
- Scheme 3: cyanovinyl should be replaced by malonitrile
- Line 168: pi should be replaced by p
- Scheme 9: transformation 39-à 40 is not clear. What is the source of “R”?
- Line 472: term “tetracyano acceptor” is not correct. I suggest something like
tetracyanobutadienyl.
- - Line 666: what does the term "Vilsmeier linker" mean?
- Scheme 26: “R” as a reagent in the transformation of 123à124-127 is not correct.
- Scheme 27: What are “heterocyclic salts”?
- In some schemes the formulas are not correctly written e.g. CHCl3 instead of CHCl3
Response to Reviewer 1:
We thank reviewer 1 for their comments and suggestions. Please see the response to each comment below. All the changes and updates are highlighted yellow in the manuscript document.
- Lines 91-95, 448-449 units are added in eV.
- In Figure 4 Cs2O3 replaced with Cs2CO3.
- In Scheme 3 cyanovinyl replaced with malonitrile.
- In line 168 pi replaced by TT linker system.
- Scheme 9 has been updated and the synthesis steps were described in more detail in text.
- The term “tetracyano acceptor” replaced with “tetracyano moiety containing”.
- Vilsmeier linker refers to the cyclohexene linker in the structure between donor and acceptor units, which is a dialdehyde compound synthesized via Vilsmeier Haack formylation. The following part added in text to clarify it.
“The acceptor and donor units in these fluorophores are connected with a bisaldehyde 129 which is synthesized via Vilsmeier Haack formylation and referred as Vilsmeier linker.”
- R was changed as reactant in the scheme 26.
- Scheme 27 was updated showing heterocyclic salts and the final fluorophore structures clearly.
- All formulas were checked, and the corrected schemes were highlighted.
Reviewer 2 Report
Comments and Suggestions for Authors
Manuscript ID: biomolecules-3265156
Title: Roadmap for Designing Donor-Acceptor Fluorophores in UV-vis and NIR Regions: Synthesis, Optical Properties and Applications
By Guliz Ersoy and Maged Henary
The presented Review reports on the synthesis, optical properties, and applications of donor-acceptor fluorophores in the UV-vis and NIR region. However, I believe that the information presented in this review will not be of interest to a wide range of researchers from the chemical community. Therefore, I cannot recommend the manuscript for publication.
The following remarks may be addressed to the authors:
1) The title is misleading. For the donor-acceptor fluorophores, it is not clarified that it is only about those with a π-linker. Moreover, all fluorophores that are part of a conjugated system can be considered as donor-acceptor fluorescent systems, not only the compounds presented in this work with the corresponding linkers and acceptors.
2) Of significant interest are the donor-acceptor fluorescent systems or the so-called wavelength shifting chromophores based on FRET and ICT because they are capable of performing valuable ratiometric fluorescence analysis. This type of fluorophores are an integral part of the donor-acceptor fluorescence systems, but unfortunately, the authors do not present such information in their review.
3) Donor-acceptor systems based on 1,8-naphthalimides, 9-phenylxanthenes, coumarins (with one exception) and other classical fluorophores with proven effective fluorescence characteristics are absent from this review.
4) The comments about AIE seem too superficial and limited to some form of educational nature.
5) Bioimaging is inherent to molecules of various natures, not only to donor-acceptor systems. As for "autofluorescence from biomolecules", this phenomenon is easily overcome with the above-commented ratiometric analysis.
6) The "Cytotoxicity" section is completely redundant. Cytotoxicity is characteristic of specific structures and cannot be put under a common denominator.
7) The "Biosensors" section is too modest to say the least.
Author Response
Reviewer 2
The presented Review reports on the synthesis, optical properties, and applications of donor-acceptor fluorophores in the UV-vis and NIR region. However, I believe that the information presented in this review will not be of interest to a wide range of researchers from the chemical community. Therefore, I cannot recommend the manuscript for publication.
The following remarks may be addressed to the authors:
1) The title is misleading. For the donor-acceptor fluorophores, it is not clarified that it is only about those with a π-linker. Moreover, all fluorophores that are part of a conjugated system can be considered as donor-acceptor fluorescent systems.
2) Of significant interest are the donor-acceptor fluorescent systems or the so-called wavelength shifting chromophores based on FRET and ICT because they are capable of performing valuable ratiometric fluorescence analysis. This type of fluorophores are an integral part of the donor-acceptor fluorescence systems, but unfortunately, the authors do not present such information in their review.
3) Donor-acceptor systems based on 1,8-naphthalimides, 9-phenylxanthenes, coumarins (with one exception) and other classical fluorophores with proven effective fluorescence characteristics are absent from this review.
4) The comments about AIE seem too superficial and limited to some form of educational nature.
5) Bioimaging is inherent to molecules of various natures, not only to donor-acceptor systems. As for "autofluorescence from biomolecules", this phenomenon is easily overcome with the above-commented ratiometric analysis.
6) The "Cytotoxicity" section is completely redundant. Cytotoxicity is characteristic of specific structures and cannot be put under a common denominator.
7) The "Biosensors" section is too modest to say the least.
Response to Reviewer 2:
We thank reviewer 2 for their comments and suggestions. Donor acceptor fluorophores with a conjugated linker system is a high interest of a wide range of chemists and biochemists since these fluorophores are modified for certain bio-applications. Therefore, we have a great interest in exploring this area of research. However, we understand the concern of the reviewer 2 about our article being precisely focused on linkers containing systems, therefore we have added the following paragraph with additional citations, which shows that there are D-A systems without a linker, however this review focuses on the ones with a linker.
“Some examples of these systems do not have a linker unit between donor and acceptor units[2,3]. These scaffolds can be fluorescent due to their conjugated moieties. Similarly, the charge transfer occurs between donor and acceptor units upon photoexcitation, resulting in ICT[4]. However, the linker addition in between can allow a more significant number of modifications to shift the wavelengths to the near infrared region. Therefore, in this review are focusing on the donor acceptor fluorophores with linker units by presenting various examples.”
- We have modified the title as “Roadmap for Designing Donor – – Acceptor Fluorophores in UV-vis and NIR Regions: Synthesis, Optical Properties and Applications”.
- We have added some examples considering the reviewer’s suggestion about radiometric analysis of D-A fluorophores. However, we are focusing on the systems with ICT therefore, the suggested FRET systems are beyond the scope of this article.
- We mainly included the scaffolds that hasn’t been explored earlier as a part of donor- - acceptor systems. Therefore, we didn’t include some scaffolds that have been included in other review papers, specifically large number of reviews were presented focusing on coumarins and xanthene scaffolds. However, we added 3 new citations in the introduction part about reviewer’s suggestions. The references are shown below.
[2] Geraghty, C.; Wynne, C.; Elmes, R.B.P. 1,8-Naphthalimide based fluorescent sensors for enzymes. Coordination Chemistry Reviews 2021, 437, 213713, doi:https://doi.org/10.1016/j.ccr.2020.213713.
[3] Han, Q.; Wang, N.; Wang, M.; Wang, J. Donor-acceptor based enhanced electrochemiluminescence of coumarin microcrystals: Mechanism study and sensing application. Sensors and Actuators B: Chemical 2023, 393, 134296, doi:https://doi.org/10.1016/j.snb.2023.134296.
[4] Chen, C.; Fang, C. Devising Efficient Red-Shifting Strategies for Bioimaging: A Generalizable Donor-Acceptor Fluorophore Prototype. Chemistry – An Asian Journal 2020, 15, 1514-1523, doi:https://doi.org/10.1002/asia.202000175.
- We have updated the applications section. We added optical properties of selected compounds and discussed the two-photon absorption and aggregation induced emission (AIE) under that category. We discussed the bioimaging properties of selected compounds with aggregation induced emission property under the bioimaging title.
- Yes, other fluorophores can be used as bioimaging agents, and we don’t want to have a misunderstanding claiming that only donor acceptor systems are used as bioimaging agents. The scope of this review article is donor acceptor systems; therefore, we only focus on these systems for bioimaging applications.
- We understand reviewer’s comment and have changed the tittle as anticancer activity.
7) Biosensors part has been revised.
Reviewer 3 Report
Comments and Suggestions for Authors
In this manuscript, the authors present a summary of the roadmap for designing donor-acceptor fluorophores in UV-vis and NIR regions. They focus on the synthesis, optical properties and applications of various D–π–A fluorophores. This review describes aspects of donor-acceptor fluorophores in UV-vis and NIR regions are relatively comprehensive, but there are still some details that require modifications. Thus, this manuscript could be publishable with a revision by addressing the following concerns:
1. The manuscript contains numerous formatting issues, such as inconsistent first-line indentation and improper placement of images. Please correct these to meet the journal's requirements.
2. The manuscript summarizes a large number of dyes with various chemical structures; however, some structures are not well-presented. When using ChemDraw, it is advisable to utilize the structure optimization feature or manually adjust the structures for better aesthetics. Additionally, please ensure that all chemical structures and fonts are uniform in size, avoiding arbitrary scaling.
3. Some reaction conditions are not labeled correctly. For instance, on page 12, line 347, “Pd(PPh3)4” should be “Pd(PPh3)4”. There are other similar cases, please correct them.
4. Many images are unclear. Any cited images should be of high resolution.
5. The section on optical characterization data is lacking in detail. If there are existing references that include tables of optical properties, please cite them appropriately or summarize relevant optical properties in a corresponding table.
6. The application section summarizes aggregation induced emission (AIE). However, we believe AIE is a luminescence mechanism rather than an application. Please make appropriate adjustments to this section.
Author Response
Reviewer 3
In this manuscript, the authors present a summary of the roadmap for designing donor-acceptor fluorophores in UV-vis and NIR regions. They focus on the synthesis, optical properties and applications of various D–π–A fluorophores. This review describes aspects of donor-acceptor fluorophores in UV-vis and NIR regions are relatively comprehensive, but there are still some details that require modifications. Thus, this manuscript could be publishable with a revision by addressing the following concerns:
- The manuscript contains numerous formatting issues, such as inconsistent first-line indentation and improper placement of images. Please correct these to meet the journal's requirements.
- The manuscript summarizes a large number of dyes with various chemical structures; however, some structures are not well-presented. When using ChemDraw, it is advisable to utilize the structure optimization feature or manually adjust the structures for better aesthetics. Additionally, please ensure that all chemical structures and fonts are uniform in size, avoiding arbitrary scaling.
- Some reaction conditions are not labeled correctly. For instance, on page 12, line 347, “Pd(PPh3)4” should be “Pd(PPh3)4”. There are other similar cases, please correct them.
- Many images are unclear. Any cited images should be of high resolution.
- The section on optical characterization data is lacking in detail. If there are existing references that include tables of optical properties, please cite them appropriately or summarize relevant optical properties in a corresponding table.
- The application section summarizes aggregation induced emission (AIE). However, we believe AIE is a luminescence mechanism rather than an application. Please make appropriate adjustments to this section.
Response to Reviewer 3:
We thank reviewer 3 for the comments and suggestions. Please find our responses below.
- We have updated the formatting and corrected the paragraph beginnings.
- The size of the structures is adjusted with the same size and font on ChemDraw.
- We have reviewed the schemes and corrected the typos regarding chemical formulas of reagents. All changes are highlighted in yellow.
- The images used in the manuscript are directly downloaded with high quality option from the original papers and this was the best quality we were able to get.
- We have added a Table to our manuscript showing the absorbance and emission properties of each compound that has been reported in the review article.
- We divided the application section to two headings as ‘Properties of selected compounds’ and ‘Applications of selected compounds. We moved the two-photon absorption and aggregation induced emission under the optical properties section.
Reviewer 4 Report
Comments and Suggestions for Authors
Please see the attached file for the detailed comments.

Author Response
Response to Reviewer 4:
We thank reviewer 4 for their comments and suggestions. Please find the our responses below.
- The manuscript is mainly focused on introducing various pathways to synthesize donor acceptor scaffolds. The optical properties are discussed in text after the synthetic scheme explanations. To make it easier for the readers, we have added Table 1 with the absorbance and emission wavelengths of each compound discussed in the review article.
- We understand the reviewer’s comment and we have changed the applications title to ‘properties and applications. For the two-photon absorption and aggregation induced emission since these are properties of the compounds and we would like to discuss them, first we discussed the property and then added the applications part that were reported in the literature. The cytotoxicity title has been changed with ‘anticancer activity’ as well.
- Usage of donor acceptor dyes as biosensors has a significance among the potential applications, therefore we added the general scaffold for compounds 138 and 139.
- We have taken permission for reusage of items that have been taken from the literature. To make it more significant we added the citation number in the caption and highlighted the figure number in the text.
- We changed the title as “aza-BODIPY Unit as Linker and Acceptor Group”. We changed the statement as follows: “There were number of studies that reported BODIPY unit containing D-A fluorophores.”
- The typos in text and in the schemes have been corrected. All corrections in text and schemes are highlighted in yellow.
- “Conformation” corrected
- eV units added.
- Anti and anti-anti means the conformation of the two thiophene units according to the paper cited. Anti is for compounds with one thiophene linker and anti-anti is referred to the compounds with two thiophene linkers.
- The typo has been corrected. The compound numbers are 60 and 61. Compound 60 containing carboxylic acid and 61 containing carboxylate ester group. The correction has been made in the scheme and text.
- Temperature corrected to –78 °C.
- The text mentioned a range of absorbance as 743-752 nm, not two absorbance maxima. To clarify this, we changed the text as follows:
The fluorophore 109 showed emission at 743 nm in methanol, at 744 nm in acetonitrile and 752 nm in chloroform.”
- NMe3 corrected to NMe3+ for compound 109, updated in the figure caption.
- “Fluorophores 134 and 135” corrected to “fluorophores 136 and 137”.
- We revised the grammatical errors in the manuscript.
Reviewer 5 Report
Comments and Suggestions for Authors
The author reviewed the synthesis, properties, and applications of D-A type fluorophores since 2014 to understand the effect of the donor and acceptor groups on the ICT. This work could be accepted in the Biosensors after minor revision.
(1) Due to the fluorophore's environment affecting its photophysical properties, the measured conditions should be added to the article, such as QY in DCM or other solvent (Line 87), and in which solvent? (Lines 166, 238, 255, and 423).
(2) Correct the synthetic step, such as Cs₂CO₃, not Cs₂O₃ (Fig. 4), malononitrile, not dicyanovinyl (Scheme 3), red-marked "R=Br"? (Scheme 9), synthesis error for compound 45 and requited subscripts for reagents under arrows (Scheme 10), and KPF₆, not KPF₆⁻ (Scheme 14). Be sure to double-check the other Figures and Schemes.
(3) Cite and discuss briefly the newest study on D-A type AIEgens in the introduction, doi: 10.1039/D4TB01617C, 10.1039/D4TB01847H.
Author Response
Reviewer 5
The author reviewed the synthesis, properties, and applications of D-A type fluorophores since 2014 to understand the effect of the donor and acceptor groups on the ICT. This work could be accepted in the Biosensors after minor revision.
(1) Due to the fluorophore's environment affecting its photophysical properties, the measured conditions should be added to the article, such as QY in DCM or other solvent (Line 87), and in which solvent? (Lines 166, 238, 255, and 423).
(2) Correct the synthetic step, such as Cs₂CO₃, not Cs₂O₃ (Fig. 4), malononitrile, not dicyanovinyl (Scheme 3), red-marked "R=Br"? (Scheme 9), synthesis error for compound 45 and requited subscripts for reagents under arrows (Scheme 10), and KPF₆, not KPF₆⁻ (Scheme 14). Be sure to double-check the other Figures and Schemes.
(3) Cite and discuss briefly the newest study on D-A type AIEgens in the introduction, doi: 10.1039/D4TB01617C, 10.1039/D4TB01847H.
Response to Reviewer 5:
We thank reviewer 5 for their comments and suggestions, see our responses below.
- We have added the quantum yield data of selected fluorophores in Table 1 at the end.
- The synthetic schemes are checked, and the typos are corrected. All corrected schemes are highlighted in yellow.
- The suggested publications are cited and discussed in the aggregation induced emission (AIE) part as shown below.
“Fluorophores with AIE property can be designed strategically to accumulate at targeted organelles. One of the examples by Zhuang et al. reported a strategical design of fluorophores with cationic pyridinium unit to target mitochondria for reactive oxygen species (ROS) generation [88]. Based on the positional isomerism strategy employed by the researchers, it was reported that the cyano group on the fluorophores with AIE induces the ROS generation. In another study by Li et al., who reported the coumarin-based fluorophores with AIE properties for H2S sensing [89]. Not limited to the AIE, but we have discussed more applications in the following section.”
Round 2
Reviewer 2 Report
Comments and Suggestions for Authors
The authors have made only cosmetic changes and have not reflected the philosophy of the notes made in the nature of recommendations. Therefore, I continue to maintain my original position that I cannot recommend the manuscript for publication.
Reviewer 4 Report
Comments and Suggestions for Authors
The authors have addressed my comments and concerns. Therefore, I recommend the acceptance of the revised manuscript.